# Multi-Agent Domain Calibration with a Handful of Offline Data

**Tao Jiang**[1,2,3]*, **Lei Yuan**[1,2,3]*, **Lihe Li**[1,2], **Cong Guan**[1,2],
**Zongzhang Zhang**[1,2]†, **Yang Yu**[1,2,3]

[1]National Key Laboratory of Novel Software Technology, Nanjing University, Nanjing, China
[2]School of Artificial Intelligence, Nanjing University, Nanjing, China
[3]Polixir Technologies, Nanjing, China
{jiangt,yuanl,lilh,guanc}@lamda.nju.edu.cn, {zzzhang, yuy}@nju.edu.cn

## Abstract

The shift in dynamics results in significant performance degradation of policies trained in the source domain when deployed in a different target domain, posing a challenge for the practical application of reinforcement learning (RL) in real-world scenarios. Domain transfer methods aim to bridge this dynamics gap through techniques such as domain adaptation or domain calibration. While domain adaptation involves refining the policy through extensive interactions in the target domain, it may not be feasible for sensitive fields like healthcare and autonomous driving. On the other hand, offline domain calibration utilizes only static data from the target domain to adjust the physics parameters of the source domain (e.g., a simulator) to align with the target dynamics, enabling the direct deployment of the trained policy without sacrificing performance, which emerges as the most promising for policy deployment. However, existing techniques primarily rely on evolution algorithms for calibration, resulting in low sample efficiency. To tackle this issue, we propose a novel framework Madoc (**M**ulti-**a**gent **do**main **c**alibration). Firstly, we formulate a bandit RL objective to match the target trajectory distribution by learning a couple of classifiers. We then address the challenge of a large domain parameter space by modeling domain calibration as a cooperative multi-agent reinforcement learning (MARL) problem. Specifically, we utilize a Variational Autoencoder (VAE) to automatically cluster physics parameters with similar effects on the dynamics, grouping them into distinct agents. These grouped agents train calibration policies coordinately to adjust multiple parameters using MARL. Our empirical evaluation on 21 offline locomotion tasks in D4RL and NeoRL benchmarks showcases the superior performance of our method compared to strong existing offline model-based RL, offline domain calibration, and hybrid offline-and-online RL baselines.

## 1 Introduction

Reinforcement learning (RL) has gained significant traction in various fields [1], such as sequential recommendation systems [2] and robotic control [3], demonstrating tremendous potential in real-world applications. However, the inherent trial-and-error nature of RL limits its application, especially in safety-critical areas such as healthcare [4] and autonomous driving [5], as extensive interactions with the target environment can entail prohibitive costs and pose substantial safety risks. To address this problem, a range of studies have proposed collecting training samples from a surrogate source domain (e.g., simulation environment) to learn policies, which are then deployed to the downstream

---

*Equal Contribution
†Corresponding Author

target domain [6, 7]. Nonetheless, due to complex system dynamics and the characteristics of open environments, a high-fidelity simulator may not always be available [8], leading to severe dynamics shifts between the source and target domains [9]. Consequently, policies trained optimally in the source domain may fail catastrophically in the target domain. To bridge the dynamics gap, various kinds of solutions have been developed recently. Domain randomization [10] methods, for instance, randomly sample the physics parameters of the source domain and train policies across multiple simulated environments to approximate the target domain. However, as the target domain is often unknown and set in an open environment [8], these methods can also suffer from unpredictable policy degradation, which hinders further development.

Integrating the source domain with some data from the target domain offers a promising solution to the mentioned problem [11]. A class of methods, known as domain calibration, attempts to use data from the target domain as feedback to calibrate the easily obtained source domain and then transfer the policy directly to the target domain. This approach shows enormous potential when the parameters are adjusted accurately enough. Some typical methods automatically tune the physics parameters by minimizing the transition discrepancy between the source and target domains [12, 13] or by maximizing the expected return in the target domain [14]. While these methods can successfully transfer learned policies in robotics [15], they still require interaction feedback from the target domain during training. Instead, DROID [16] and DROPO [17] introduce an offline setting for domain calibration, where the physics parameters are adjusted using offline demonstrations pre-collected in the target domain, showing potential for real-world applications.

Nevertheless, in complex real-world scenarios, numerous physics parameters may require calibration. The above-mentioned methods primarily employ evolutionary algorithms [16, 18] or sampling-based methods [19, 20] for black-box optimization, often results in low sample efficiency [21, 22]. Recently, some algorithms have attempted to mitigate this issue by learning sampling strategies [22] or leveraging causal discovery [23] to eliminate parameters that have little impact on the environment. Despite the effectiveness of these methods, a significant challenge remains in handling complex scenarios where all physics parameters critically influence the dynamics, and different parameters may have varying, or even opposite, impacts on these dynamics [24]. A method for efficiently addressing the interrelations among different parameters is urgently needed.

From the perspective of whole-domain calibration, each physics parameter contributes to different aspects of the calibration process. This can be modeled as a typical multi-agent system (MAS) problem [25], where each agent adjusts a group of domain parameters, and all agents cooperate to reduce the domain gap. This problem can be addressed using cooperative multi-agent reinforcement learning (MARL) [26], leading to the development of the Madoc (**M**ulti-**a**gent **do**main **c**alibration) framework. Specifically, we first formulate domain calibration as a target trajectory distribution matching problem and derive a bandit RL optimization objective by introducing a couple of classifiers to act as the reward model. We then formulate the problem into the MAS where multiple agents calibrate different parameters to reduce the dynamics gap between the source and target domains. Concretely, we propose an automatic grouping technique to cluster physics parameters based on their impacts on the dynamics. We then employ popular value decomposition methods in MARL to train cooperative calibration policies to adjust domain parameters. We conduct experiments on popular locomotion tasks to showcase Madoc's superior performance against baselines and highlight the contributions of its core design components. The source code is available at `https://github.com/LAMDA-RL/Madoc`.

## 2   Related Work

**Domain transfer in RL.** Transferring RL policies learned from imperfect source domains to the target domain is a crucial step in the practical use of RL algorithms [11]. However, the trained policy often suffers from severe performance deterioration when directly deployed into the target domain due to the distribution shift between different domains with varying transition dynamics [27]. Previous works have addressed this problem with three common strategies: domain randomization (DR), domain adaptation (DA), and system identification (SI). DR attempts to train a generalizable policy that works well across a variety of randomized simulated dynamics [28, 9]. While the motivation is simple and often effective, these methods require manually determining which parameters to randomize and may result in underfitting or failing policies due to hand-tuning parameter ranges. DA involves using a huge amount of data from source domains to improve policy performance on a

different target domain [29, 30, 31]. However, these efforts are constrained by the quality and quantity of target domain data and often still require interaction with the target domain. Another line of work, SI, uses measured data to build mathematical models of dynamical systems [32]. These methods rely on numerous interactions with the target domain to study how to learn a model of the system dynamics [33, 34], which results in learning a biased policy with fewer interactions. Most recent works calibrate the parameters of the biased source domain to bridge the domain gap [19, 13, 15], also known as domain calibration, and try to improve efficiency by learning a parameter sampling strategy [22] or leveraging causal discovery [23]. However, these methods still require interacting with the target domain, posing potential safety hazards during the training process. To mitigate this problem, DROID [16] and DROPO [17] use offline datasets to adjust the source domain parameters via evolutionary algorithms [18] with different optimization objectives, perform poorly when the number of domain parameters is large [35]. Unlike the above methods, we propose to adjust the source domain with a handful of offline data, enabling the domain parameters to match the target trajectory distribution with high sample efficiency.

**Cooperative multi-agent RL.** Many real-world problems are inherently large-scale and complex, making it inefficient and impractical to model them as single-agent systems. Instead, they are more suitably addressed as multi-agent systems (MASs) [25]. Multi-agent reinforcement learning (MARL) provides frameworks for modeling and solving such challenges [26]. In scenarios where agents within MAS share common objectives, these problems are categorized under cooperative MARL, which has demonstrated significant advancements in domains like power management [36], path planning [37], and dynamic algorithm configuration [38]. One of the primary challenges in cooperative MARL is the scalability issue [39, 40, 41], exacerbated by the exponential growth of the search space with the number of agents, complicating policy exploration and learning. Various approaches have been proposed to enhance agent coordination recently. These include policy-based methods such as MADDPG [42] and MAPPO [43], value-based techniques like VDN [44] and QMIX [45], and innovations like the transformer architecture [46]. Among these methods, value-based approaches have demonstrated promising results in diverse and complex settings [47, 48]. VDN leverages additivity to factorize global value functions, QMIX further enforces monotonicity in global value functions, and DOP [49] introduces value function decomposition within multi-agent actor-critic frameworks. These methods exhibit remarkable coordination capabilities across various tasks such as SMAC, Hanabi, and GRF [26]. In this paper, our method formulates domain calibration as a cooperative MARL problem, improving efficiency and fidelity.

## 3 Background

**Reinforcement Learning** can be generally modeled as a Markov decision process (MDP) [50], formulated as a tuple $\mathcal{M} := (\mathcal{S}, \mathcal{A}, T, r, \gamma, \rho_0)$, where $\mathcal{S}$ and $\mathcal{A}$ denote the state and action spaces, $T(s'|s, a) \in [0, 1]$ and $r(s, a)$ represent the transition and reward functions, $\gamma \in [0, 1)$ implies the discount factor, and $\rho_0(s)$ is the initial state distribution. The agent running in the environment perceives the state $s_t \in \mathcal{S}$ at time step $t$, performs an action $a_t \in \mathcal{A}$ based on a learnable policy $\pi(a|s) \in [0, 1]$, then the environment receives the action, transits to a new state $s_{t+1}$, and rewards the agent according to the transition function $T(s_{t+1}|s_t, a_t)$ and reward function $r(s_t, a_t)$ at next time step. The above process is continuously iterated until termination, we can record the whole trajectory of length $H + 1$ as $\tau = (s_0, a_0, r_0, s_1, a_1, r_1, \cdots, s_H, a_H, r_H)$ and the trajectory distribution over agent's policy and the environment can be defined as $d_{\pi,\mathcal{M}}(\tau) = \rho_0(s_0) \prod_{t=0}^{H} T(s_{t+1}|s_t, a_t)\pi(a_t|s_t)$. The objective of RL algorithms is to learn a policy $\pi(a|s)$ which maximizes the expected discounted return across the distribution of trajectories, i.e., $\mathcal{J}(\mathcal{M}, \pi) = \mathbb{E}_{\tau \sim d_{\pi,\mathcal{M}}(\cdot)} R(\tau)$ with $R(\tau) = \sum_{t=0}^{H} \gamma^t r(s_t, a_t)$.

**Domain Calibration** aims to adjust a manipulable source domain to close the domain gap between it and the target domain. Both the target and source domains can be modeled as MDPs, and the only difference between them is the transition functions which are determined by the physics dynamics parameter vector $\xi \in \Xi \subset \mathbb{R}^N$ (e.g., friction, mass, damping). Here, $\Xi$ denotes the physics parameter space, and $N$ represents the dimension of the physics parameters. Each parameter $\xi^i$ is bounded on a closed interval, which can only be inferred roughly based on experience and expert knowledge, i.e., $\xi^i \in [\xi^i_{\text{low}}, \xi^i_{\text{high}}]$. We assume the unknown physics parameters of the target domain $\xi^*$ are included in the parameter space $\Xi$ if the physical modeling is reasonable, i.e., $\xi^* \in \Xi$, as we can set sufficient wide parameter ranges. We now denote the transition function conditioned on domain parameters as $T_\xi = T(s'|s, a, \xi)$ and the corresponding MDP as $M_\xi := (\mathcal{S}, \mathcal{A}, T_\xi, r, \gamma, \rho_0)$. However, the

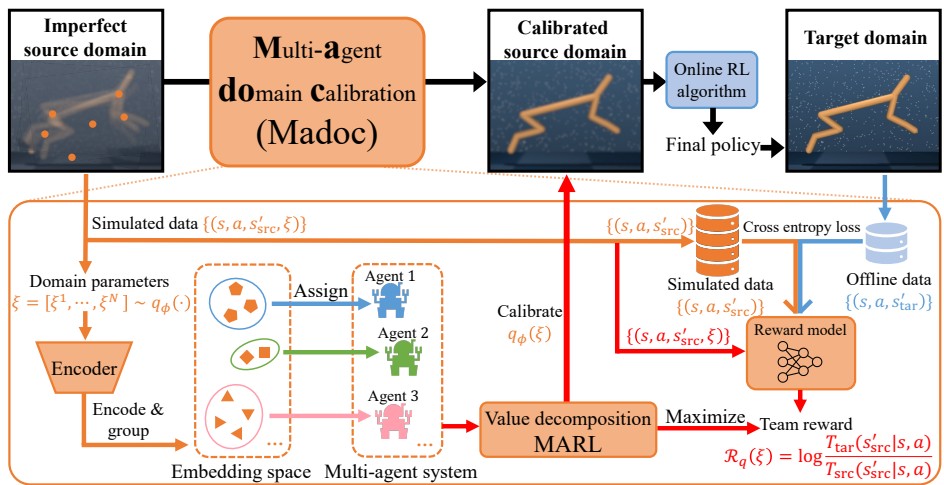

Figure 1: The conceptual workflow of the multi-agent domain calibration framework. The orange arrow represents the simulated data flow in the source domain, with the transition function $T(s'|s, a, \xi)$, while the blue represents the offline data in the target domain, with the transition function $T(s'|s, a, \xi^*)$. The subscripts "src" and "tar" are used to distinguish between the source and target domains, respectively. After learning the grouping scheme, we use the red arrow to represent the process of domain calibration by MARL value decomposition methods.

manipulable source domain is typically non-differentiable, we can only calibrate the distribution of the source domain parameters $\xi \sim q_\phi(\cdot)$, and the optimal policy learned under this distribution is marked as $\pi^*(q_\phi) = \arg\max_\pi \mathbb{E}_{\xi \sim q_\phi(\cdot)} \mathcal{J}(M_\xi, \pi)$. The objective of domain calibration is to learn the source domain parameters that maximize the expected discounted return under the target domain: $\max_{q_\phi} \mathcal{J}(M_{\xi^*}, \pi^*(q_\phi))$.

## 4 Method

In this section, we propose the Madoc (**M**ulti-**a**gent **do**main **c**alibration) framework for leveraging a modest amount of offline data from the target domain to calibrate the biased source domain, thus facilitating optimal policy transfer. The overall workflow of the Madoc framework is shown in Fig. 1. We first deduce a bandit RL objective to adjust the domain parameters in Sec. 4.1, improving the synthetic data sampled from the source domain to align with the target trajectory distribution. We further model it as a cooperative multi-agent reinforcement learning problem in Sec. 4.2 and use an automatic grouping technique to improve the efficiency and fidelity of domain calibration. Finally, a practical algorithm under the Madoc framework is presented in Sec. 4.3.

### 4.1 Domain Calibration via Reinforcement Learning

Domain calibration is the process of tuning the parameter distribution of a mismatched source domain to better align with the target domain, which can be realized by comparing the divergence between target domain interactions and simulated synthetic rollouts based on the same policy [12, 13]. However, since interacting with the target domain may not be feasible for sensitive fields, we propose an alternative approach to minimize the trajectory discrepancy between the two domains by employing a handful of offline target domain data.

Formally speaking, the static offline dataset $\mathcal{D} = \{\tau_1, \tau_2, \cdots, \tau_k\}$ contains $k$ trajectories where $\tau_i = (s_0^i, a_0^i, r_0^i, s_1^i, a_1^i, r_1^i, \cdots, s_H^i, a_H^i, r_H^i)$, which are collected previously by an unknown behavior policy $\mu$ from the target domain, i.e., $\tau_i \sim d_{\mu, \mathcal{M}_{\xi^*}}(\cdot)$. By introducing a prior normal parameter distribution $p(\xi)$ to foster better generalization [51], we intend to learn a sample policy $\pi$ and calibrate the domain parameters to match the target trajectory distribution:

$$\min_{\pi, q_\phi} D_{\mathrm{KL}}\left(q_\phi(\xi) d_{\pi, \mathcal{M}_\xi}(\tau) || p(\xi) d_{\mu, \mathcal{M}_{\xi^*}}(\tau)\right), \tag{1}$$

where the Kullback-Leibler (KL) divergence can be further derived as:

$$\mathbb{E}_{\substack{\xi \sim q_\phi(\cdot) \\ \tau \sim d_{\pi,\mathcal{M}_\xi}(\cdot)}} \left[ \log \frac{\prod_{t=0}^H \pi(a_t|s_t) T(s_{t+1}|s_t, a_t, \xi)}{\prod_{t=0}^H \mu(a_t|s_t) T(s_{t+1}|s_t, a_t, \xi^*)} + \log \frac{q_\phi(\xi)}{p(\xi)} \right], \tag{2}$$

$$= \mathbb{E}_{\substack{\xi \sim q_\phi(\cdot) \\ \tau \sim d_{\pi,\mathcal{M}_\xi}(\cdot)}} \left[ \sum_{t=0}^H \left( \log \frac{\pi(a_t|s_t)}{\mu(a_t|s_t)} + \log \frac{T(s_{t+1}|s_t, a_t, \xi)}{T(s_{t+1}|s_t, a_t, \xi^*)} \right) \right] + \mathbb{E}_{\xi \sim q_\phi(\cdot)} \left[ \log \frac{q_\phi(\xi)}{p(\xi)} \right], \tag{3}$$

$$\approx \mathbb{E}_{(s,a) \sim \mathcal{B}} \left[ \log \frac{\pi(a|s)}{\mu(a|s)} \right] - \mathbb{E}_{(s,a,s',\xi) \sim \mathcal{B}} \left[ \log \frac{T(s'|s,a,\xi^*)}{T(s'|s,a,\xi)} \right] + D_{\mathrm{KL}}(q_\phi(\xi) \| p(\xi)), \tag{4}$$

where Eq. 4 is an approximation of Eq. 3 as the parameter distribution and trajectory distribution used to calculate the expectation are difficult to compute. Consequently, we use Monte Carlo sampling on the source domain to approximate the expected results. To enhance sampling efficiency, we sample a domain parameter $\xi \sim q_\phi(\cdot)$, generate the trajectories $\tau \sim d_{\pi,\mathcal{M}_\xi}(\cdot)$, and store the rollouts $(s, a, s', \xi)$ in the replay buffer $\mathcal{B}$. By doing so, we are able to convert the trajectory-based objective into a transition-based one, following the classic off-policy RL paradigm.

It is delighted to discover that the objective in Eq. 4 can be clearly divided into three terms: the first term, i.e., $\min_\pi \mathbb{E}_{(s,a) \sim \mathcal{B}} \left[ \log \frac{\pi(a|s)}{\mu(a|s)} \right] \approx \min_\pi D_{\mathrm{KL}}(\pi(a|s) \| \mu(a|s))$, attempts to minimize the KL divergence between $\pi(a|s)$ and $\mu(a|s)$, we can consider it as a variant of behavior cloning; the second term is formulated as $\max_{q_\phi} \mathbb{E}_{(s,a,s',\xi) \sim \mathcal{B}} \left[ \log \frac{T(s'|s,a,\xi^*)}{T(s'|s,a,\xi)} \right]$, which can be seen as a bandit RL objective for policy $q_\phi(\xi)$ to maximize reward $\mathcal{R}_q(\xi) = \log \frac{T(s'|s,a,\xi^*)}{T(s'|s,a,\xi)}$; and the last term is regarded as a policy regularizer added on $q_\phi(\xi)$ to prevent it from collapsing. It is worth noting that the policy $q_\phi(\xi)$ here is not the one running (sampling) on the source domain, but the one outputting physics parameter vector $\xi$ as an action to adjust the source domain. To prevent confusion, in the following paper, the policy running on the source domain, i.e., $\pi(a|s)$, is referred to as the "running policy", while the one adjusting the domain parameters, i.e., $q_\phi(\xi)$, is referred to as the "calibration policy". Additionally, we similarly define the "calibration critic", which is responsible for evaluating the accuracy of the parameters output by the calibration actor. The calibration critic and the calibration policy (actor) together constitute a calibration agent.

The key challenge lies in how to estimate the stochastic reward $\log \frac{T(s'|s,a,\xi^*)}{T(s'|s,a,\xi)}$ given the offline data and simulated rollouts. According to Bayes' rule, we can transform the transition probability [29] as:

$$T(s'|s,a,\xi) = \frac{P(\xi|s,a,s') P(s,a,s')}{P(\xi) P(s,a|\xi)} = \frac{P(\xi|s,a,s') P(s,a,s')}{P(\xi|s,a) P(s,a)}, \tag{5}$$

and the reward can be derived as:

$$\mathcal{R}_q(\xi) = \log P(\xi^*|s,a,s') - \log P(\xi^*|s,a) - \log P(\xi|s,a,s') + \log P(\xi|s,a),$$

where $\xi^*$ and $\xi$ stand for the target and source domains respectively. Hence, we can train a couple of binary classifiers $D_{\psi_{\mathrm{sas}}}(\cdot|s,a,s')$ and $D_{\psi_{\mathrm{sa}}}(\cdot|s,a)$ to discriminate whether state-action-state and state-action pairs come from the offline dataset (referred to as the binary variable $\mathrm{target}$) or synthetic samples (referred to as the binary variable $\mathrm{source}$). These two discriminators form a reward model with certain generalization ability [52], and the corresponding cross-entropy losses are written as:

$$\mathcal{L}_{\psi_{\mathrm{sas}}} = -\mathbb{E}_{(s,a,s') \sim \mathcal{D}}[\log D_{\psi_{\mathrm{sas}}}(\mathrm{target}|s,a,s')] - \mathbb{E}_{(s,a,s') \sim \mathcal{B}}[\log D_{\psi_{\mathrm{sas}}}(\mathrm{source}|s,a,s')],$$
$$\mathcal{L}_{\psi_{\mathrm{sa}}} = -\mathbb{E}_{(s,a) \sim \mathcal{D}}[\log D_{\psi_{\mathrm{sa}}}(\mathrm{target}|s,a)] - \mathbb{E}_{(s,a) \sim \mathcal{B}}[\log D_{\psi_{\mathrm{sa}}}(\mathrm{source}|s,a)]. \tag{6}$$

## 4.2 Multi-Agent Domain Calibration

As the complexity of the source domain grows, characterized by an expanding number of physics parameters, the calibration policy learned by single-agent reinforcement learning often struggles to consistently reduce the domain gap. To address this challenge, we employ multi-agent reinforcement learning (MARL) algorithms to effectively reduce the search space for a single calibration agent, thereby enhancing the efficiency and fidelity of domain calibration.

We conduct experiments on the *HalfChee-tah* [53] environment and train to calibrate the *gravity* coefficient along with other physics parameters. The preliminary results are shown in Fig. 2, we capture snapshots for both approaches at the same training step to investigate Pearson correlation [54] between the critic value of the *gravity* coefficient and the absolute calibration error. Here, the absolute calibration error represents the absolute difference between the parameters output by the calibration actor and the target parameters. The single-agent method, represented by the blue dots, utilizes just a shared calibration critic for parameter adjusting, facing challenges in assessing a specific action within a huge action space. In contrast, the multi-agent method, depicted by the red dots, employs value decomposition algorithms [44] to narrow the action space of each individual calibration policy. This decomposition leads to more accurate evaluations of domain parameters, thereby reducing calibration errors and improving policy transfer.

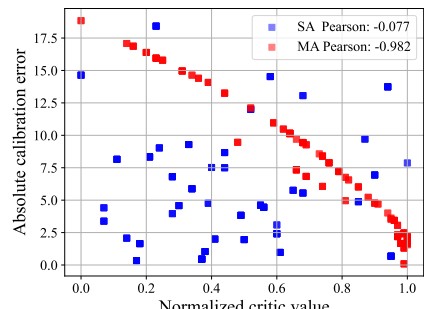

Figure 2: The Pearson correlation between the critic value of the *gravity* coefficient and the absolute calibration error. Each dot represents a sampled action, which is then fed into the corresponding critic to compute the critic value. When the parameters output by the calibration actor are closer to the target parameters (indicating a smaller absolute calibration error), the evaluation value output by a "good" critic should be higher.

To utilize the cooperative MARL for domain calibration, we now formally define $N$ agents (since the source domain has a total of $N$ physics parameters) to perform domain calibration coordinately where each of them $q_\phi^i(\xi^i)$ attempts to adjust the single parameter $\xi^i$, therefore the joint calibration policy can be decomposed as $q_\phi(\xi) = \prod_{i=1}^N q_\phi^i(\xi^i)$. Utilizing any off-the-shelf value decomposition algorithm VD, we employ the single global rewards $[\mathcal{R}_q^1(\xi^i), \cdots, \mathcal{R}_q^N(\xi^N)] = \text{VD}(\mathcal{R}_q(\xi))$ to guide individual calibration policy updates:

$$\max_{q_\phi^i} \mathbb{E}_{\xi^i \sim q_\phi^i(\cdot)} \mathcal{R}_q^i(\xi^i) - D_{\text{KL}}\left(q_\phi^i(\xi^i) \| p^i(\xi^i)\right). \tag{7}$$

Nonetheless, when the number of the domain parameters $N$ is large, employing $N$ agents for domain calibration leads to low exploration and optimization efficiency [55]. To mitigate this problem, clustering physics parameters with similar effects on the transition dynamics, e.g., the mass of a symmetrical robot's left and right feet, into one calibration agent is an advisable choice. Hence we introduce an automatic grouping technique by adopting a Variational Autoencoder (VAE) [56]. By adjusting a specific parameter $\xi^i$ while keeping all other parameters $\xi^{-i}$ fixed [57], we can assume the identity $i$ of each parameter (i.e., the one-hot encoding) to be representative of the transition $\text{tr} = (s, a, s', \xi^i)$, the Evidence Lower BOund (ELBO) of the transition is then derived as:

$$\log P(\text{tr}) \geq \mathbb{E}_{z \sim f_e(z|i)}[\log f_d(\text{tr}|z)] - D_{\text{KL}}\left(f_e(z|i) \| p(z)\right), \tag{8}$$

where $f_e, f_d$ stand for the encoder and decoder, $z$ is the latent variable, and $p(z)$ represents the corresponding prior distribution. We can deduce the reconstruction term as:

$$\begin{aligned} \log f_d(\text{tr}|z) &= \log\left[f_d(s'|s, a, \xi^i, z) f_d(s, a, \xi^i|z)\right], \\ &= \log f_d(s'|s, a, \xi^i, z) + c, \end{aligned} \tag{9}$$

where $c$ is a constant as $s$, $a$, and $\xi^i$ do not depend on the latent variable $z$. We parameterize the encoder $f_e$ and the decoder $f_d$ with $\Psi$, making them $f_{\Psi_e}$ and $f_{\Psi_d}$, and optimize the VAE model with the following loss:

$$\mathcal{L}_\Psi = -\mathbb{E}_{(s,a,s',\xi^i,\xi^{-i}) \sim \mathcal{B}, z \sim f_{\Psi_e}(\cdot|i)}\left[\log f_{\Psi_d}(s'|s, a, \xi^i, z)\right] + D_{\text{KL}}\left(f_{\Psi_e}(z|i) \| p(z)\right). \tag{10}$$

Before domain calibration, we first minimize the VAE loss (Eq. 10). Then we apply the k-means clustering method [58] to the means generated by the encoder $f_{\Psi_e}(z|i)$ for all $i \in N$, in order to group the domain parameters [39]. The resulting $n$ $(1 \leq n \leq N)$ grouping scheme is recorded as $[\xi^{g1}, \cdots, \xi^{gn}]$, each group forms one agent equipped with a calibration actor $q_\phi^{gi}(\xi^{gi})$ and a calibration critic $v_\Phi^{gi}(\xi^{gi})$, following the multi-agent actor-critic framework.

### 4.3 Practical Algorithm

Based on the above analysis, we now present a practical algorithm under the Madoc framework, the pseudo-code is shown in App. A. We apply DOP [49], a popular multi-agent policy gradient algorithm as the value decomposition method. Concretely, there are $n$ agents for calibrating the source domain after automatic grouping, we factor the joint critic as a weighted summation of individual critics:

$$V_\Phi^{\text{tot}} = \sum_{i=1}^{n} k_i v_\Phi^{gi}(\xi^{gi}) + b, \tag{11}$$

where $k_i \geq 0$ and $b$ are denoted as learnable weights and biases. The individual critics are learned by back-propagating gradients from global Temporal Difference updates:

$$\mathcal{L}_\Phi = \mathbb{E}_{(s,a,s',\xi)\sim\mathcal{B}} \left[ \frac{1}{2} \left( V_\Phi^{\text{tot}} - \mathcal{R}_q(\xi) \right)^2 \right]. \tag{12}$$

Given individual critics, we use SAC [59] to update the stochastic actors in an off-policy manner:

$$\mathcal{L}_\phi = \mathbb{E}_{\xi^{gi}\sim q_\phi^{gi}(\cdot)} \left[ \alpha \log q_\phi^{gi}(\xi^{gi}) - v_\Phi^{gi}(\xi^{gi}) + \lambda D_{\text{KL}} \left( q_\phi^{gi}(\xi^{gi}) || p^{gi}(\xi^{gi}) \right) \right], \tag{13}$$

where $\alpha$ and $\lambda$ control the relative importance of the entropy and regularization term respectively. Besides, the prior domain parameter distribution $p^{gi}(\xi^{gi})$ is set to be an exponentially moving average of the current calibration policy $q_\phi^{gi}(\xi^{gi})$, which has been shown to stabilize training like target network. Finally, we parameterize the policy running in the source domain $\pi$ with $\theta$ and enable it to clone the behavior policy on offline data during domain calibration. Once domain calibration is complete, we train the policy $\pi_\theta$ from scratch on the source domain using SAC, and directly deploy it to the target domain.

## 5 Experiments

In this section, we present the empirical evaluations of our proposed Madoc framework. We first describe the experiment environments and related baselines in Sec. 5.1, and then conduct a series of experiments to answer the following questions: (1) How is the comprehensive performance of Madoc against multiple baselines (Sec. 5.2)? (2) How do core components of Madoc contribute to the overall performance (Sec. 5.3)? (3) How is the generalization capability of Madoc across datasets of varying sizes and source domains with different initial ranges (Sec. 5.4)?

### 5.1 Experiment Setup

In our experiments, we evaluate Madoc on classic continuous control tasks from the MuJoCo [53] engine and choose two offline benchmarks to serve as offline datasets collected in the target domain. On the popular D4RL benchmark [60], we choose four locomotion tasks (*HalfCheetah*, *Hopper*, *Walker2d*, *Ant*), each with three types of datasets (*medium*, *medium-replay*, *medium-expert*), to evaluate different algorithms' performance when faced with datasets of varying quality. Considering more challenging scenarios, three environments (*HalfCheetah*, *Hopper*, *Walker2d*) along with three levels of datasets (*low*, *medium*, *high*) from NeoRL benchmark [61] are also selected. The main difference between the two benchmarks lies in that the static datasets in the NeoRL benchmark occupy more narrow distributions. During the training process, we are given imperfect source domains, and only aware of the initial range of specific physics parameters (*gravity*, *body_mass*, *dof_damping*). Each environment has different parameter dimensions, initial ranges, and ground truth values, refer to App. D for detailed information.

Madoc utilizes static offline datasets to calibrate biased source domains, and we choose several baseline algorithms with identical or similar settings for comparison. DROPO [17] and DROID [16] use fixed offline datasets from the target domain to optimize the distribution bounds with different objectives via evolution algorithms like CMA-ES [18]. OTED [51] models the parameter optimization process as a bandit RL problem, which is similar to our method, but the objective is different and cannot cope with large parameter space. H2O [62] and DR+BC are two hybrid offline-and-online algorithms where the former penalizes the Q-function learning on simulated state-action pairs with large dynamics gaps, and the latter directly combines uniform domain randomization with behavior

Table 1: Normalized average returns on D4RL benchmark. The results are evaluated in the target domain and we **bold** the highest mean.

| Task | DROPO | DROID | OTED | H2O | DR+BC | CQL | MOREC | Madoc-S | Madoc |
|------|-------|-------|------|-----|-------|-----|-------|---------|-------|
| hfctah-med | 41.0± 7.9 | 29.9± 7.5 | 76.1±11.7 | 57.3± 3.7 | 31.9±14.5 | 52.0± 3.0 | 73.9± 3.0 | **93.3± 9.4** | 91.9± 7.7 |
| hfctah-med-rep | 51.2±16.2 | 45.3± 8.0 | 67.5±13.4 | 50.4± 3.7 | 38.6± 4.4 | 48.9± 2.8 | 74.1± 2.8 | 84.7±18.6 | **95.7± 9.9** |
| hfctah-med-exp | 55.4±10.4 | 46.9±11.3 | 78.9±12.9 | 55.3± 4.1 | 40.5± 4.1 | 53.0± 2.9 | 72.0± 3.1 | 70.7±23.8 | **96.9± 5.3** |
| hopper-med | 59.1±34.6 | 73.9±10.4 | 49.8±21.4 | 83.7±19.0 | 43.2±24.7 | 50.3±18.8 | **105.0± 1.0** | 57.5±17.0 | 76.0±13.9 |
| hopper-med-rep | 43.0±18.7 | 45.3±17.6 | 65.4±26.1 | 84.1±12.3 | 43.5±16.9 | 70.0±13.9 | 28.7± 0.7 | 79.9±34.1 | **90.2±11.7** |
| hopper-med-exp | 80.4±21.3 | 21.5± 8.3 | 41.1±20.7 | 89.0±11.3 | 63.1±24.0 | 68.5±12.1 | **106.0± 0.8** | 47.7±13.0 | 81.5±18.6 |
| walker-med | 61.5±21.1 | 63.2±12.1 | 58.8±31.6 | 75.5± 8.7 | 57.2±12.1 | 4.5± 3.5 | 84.1± 0.8 | 69.9±19.8 | **90.5±17.5** |
| walker-med-rep | 19.8±16.6 | 16.8± 8.7 | 71.2±22.5 | 83.4± 1.3 | 43.7± 5.5 | 62.4±13.1 | **85.4± 0.3** | 60.6±33.1 | 85.8±20.8 |
| walker-med-exp | 60.0±13.8 | 73.8± 9.6 | 74.8±28.2 | **91.7± 7.7** | 61.1± 7.3 | 12.2± 8.3 | 86.2± 0.5 | 60.7±18.3 | 79.9±12.8 |
| ant-med | 16.4±12.2 | 20.8±17.8 | 65.3±41.8 | 60.0±26.6 | 29.2±12.6 | 58.0±20.6 | 64.9±41.2 | 76.5±29.6 | **88.7±24.8** |
| ant-med-rep | 64.1±31.9 | 64.4±35.0 | 62.4±41.9 | **98.4±12.7** | 34.8±15.0 | 43.8±33.7 | 6.1±14.1 | 58.8±42.9 | 81.2±16.5 |
| ant-med-exp | 76.7±34.0 | 64.2±41.1 | 70.0±35.5 | 66.5±22.8 | 30.3± 9.2 | 14.4±19.6 | 67.8±35.6 | 65.6±24.5 | **101.0±21.5** |
| Average | 52.4 | 47.2 | 65.1 | 74.6 | 43.1 | 44.8 | 71.2 | 68.9 | **88.3** |

Table 2: Normalized average returns on NeoRL benchmark. The results are evaluated in the target domain and we **bold** the highest mean.

| Task | DROPO | DROID | OTED | H2O | DR+BC | CQL | MOREC | Madoc-S | Madoc |
|------|-------|-------|------|-----|-------|-----|-------|---------|-------|
| HalfCheetah-L | 49.5±16.8 | 64.8±21.3 | 60.9± 8.7 | 33.1± 3.0 | 31.3± 3.1 | 32.0± 2.9 | 50.6± 0.3 | 63.2±22.7 | **71.9±10.8** |
| HalfCheetah-M | 33.6± 9.8 | 30.5± 8.1 | 64.6±15.9 | 37.1±13.7 | 34.8± 4.4 | 51.9± 4.8 | 1.1± 0.6 | 63.9±18.0 | **85.6± 7.3** |
| HalfCheetah-H | 45.4±16.6 | 47.1±17.5 | 61.0±15.4 | 2.5± 3.4 | 42.5±10.4 | 30.6±25.5 | 58.6±22.9 | 80.1±10.5 | **84.2± 6.0** |
| Hopper-L | 62.6±17.0 | 37.8±14.0 | 39.8±19.8 | 24.1± 2.0 | 40.3±11.5 | 20.7± 8.1 | 25.6± 0.7 | 56.9±16.5 | **71.7±16.9** |
| Hopper-M | **67.2±11.9** | 61.3±18.1 | 50.4±16.5 | 39.2± 6.0 | 49.4±16.4 | 35.1±10.4 | 47.2±13.4 | 47.2±13.4 | 61.4±23.8 |
| Hopper-H | 56.6±25.9 | 27.1±14.9 | 29.4±16.1 | 55.2± 6.5 | 42.4±15.7 | 55.1±14.1 | 40.6± 3.0 | 56.0±24.4 | 62.6±18.1 |
| Walker2d-L | 25.8± 8.5 | 9.9± 2.9 | 35.1±20.9 | 47.3± 3.3 | 34.1±14.0 | 33.8± 2.9 | 41.0±25.1 | 48.7±24.5 | **62.6±13.0** |
| Walker2d-M | 16.8± 6.2 | 44.8±17.4 | 53.1±21.9 | 44.3± 5.3 | 37.0± 4.5 | 57.4± 3.1 | **78.1± 1.1** | 47.5±33.1 | 69.6±20.5 |
| Walker2d-H | 20.9±10.1 | 11.9± 3.8 | 41.7±26.7 | 36.0± 7.9 | 37.8± 9.7 | **77.1± 5.7** | 20.3±30.7 | 50.9±22.0 | 67.9±18.7 |
| Average | 42.0 | 37.2 | 48.4 | 35.4 | 38.8 | 43.7 | 41.1 | 57.2 | **70.8** |

cloning. CQL [63] is the classic pure offline RL algorithm using value regularization, MOREC [52] is the state-of-the-art offline model-based RL algorithm by learning a generalizable dynamics reward function, representing the upper-bound performance by solely utilizing offline data. Madoc-S is an ablation algorithm of our proposed Madoc, which only uses a single calibration agent for parameter tuning, without applying the multi-agent decomposition and automatic grouping technique. Madoc and all the above baselines use SAC [59] as the online algorithm after domain calibration for its practicality and convenience on MuJoCo tasks.

All the numerical results in our experiments are reported in terms of mean and 95% confidence interval computed over 6 random seeds, and we give more details about the model implementation of our method in App. C.

## 5.2 Performance Comparison on the Benchmarks

We first evaluate the overall performance of our method and baselines on the two benchmarks and use the normalized average scores introduced by D4RL [60] for intuitive comparison. In realistic scenarios, both the size of offline datasets and the usage of the source domain can be restricted, thus we constrain each algorithm to access a static dataset with up to $2 \times 10^5$ transitions and execute only $1 \times 10^6$ simulated samples for domain calibration (when used). Due to these constraints, the performance of the baseline algorithms has experienced varying degrees of degradation. As shown in Tab. 1 and Tab. 2, we observe that our methods outperform prior methods on both benchmarks: two pure offline algorithms, CQL and MOREC, suffer from severe performance degradation in some scenarios, especially when encountering narrow NeoRL datasets; two hybrid offline-and-online algorithms are limited by the dataset quality and domain fidelity, and cannot achieve the expected performance; DROPO and DROID employ evolutionary algorithms to optimize domain parameters, encountering challenges in transferring learned policies when the parameter space is large; the comparison with OTED demonstrates the rationality of our optimization objective described in Sec. 4.1, while the comparison with Madoc-S further illustrates the effectiveness of modeling domain calibration as a multi-agent system formulated in Sec. 4.2. To verify that Madoc indeed reduces the dynamics gap between the source and target domains, we report the mean absolute calibration error for different algorithms in App. E.1. Furthermore, given the significant variance in domain

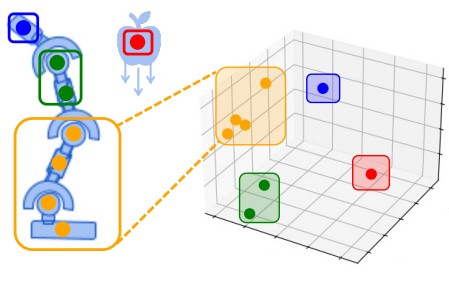 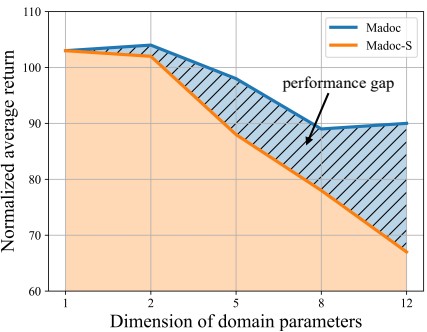

| (a) Visualization of the automatic grouping technique | (b) The performance gap between Madoc and Madoc-S |

Figure 3: (a) The visualization results of the automatic grouping technique. The left part is a schematic of the *Hopper* robot, each point on it representing a physics parameter to be calibrated, and different colors indicate the final grouping results. For example, the four parameters encircled by the yellow rectangle are clustered into one group in the embedding space on the right part. (b) The normalized average return of Madoc and Madoc-S on the *Ant* environment. We can observe that as the parameter dimension of the source domain increases, the performance gap between them (indicated by the black shadow) becomes more pronounced.

calibration algorithms, we provide a more detailed discussion on the stability of the experimental results in App. E.3.

## 5.3 Effectiveness of Different Components

To investigate the impact of the design components of Madoc, we first design experiments on the *Hopper* environment to visualize the automatic grouping technique, as shown in Fig. 3(a). The *Hopper* is a one-legged robot simulation with four distinct body sections: the torso, thigh, leg, and foot. These components are connected by three joints, which serve as the articulation points between each pair of bodies. We need to calibrate the *gravity* coefficient, the *mass* of each body, and the *damping* coefficients at each joint. After projecting the embedding space onto a three-dimensional space, we can discover that the robot's physics parameters are divided in an orderly manner from top to bottom, while the *gravity* coefficient forms a separate group on its own. The result aligns perfectly with our expectations, as parameters that have similar impacts on dynamics can be adjusted using one calibration policy. Additionally, in Fig. 3(b), we conduct ablation studies on the *Ant* environment of the D4RL benchmark, to verify the effectiveness of modeling domain calibration as an MARL problem. When the dimension of the domain parameters is 1 or 2, there is no significant performance gap between Madoc and Madoc-S; however, as the source domain becomes more complex, the performance of both declines, but the drop is faster when only one agent is employed for parameter calibration. This experimental result favorably supports that modeling as MARL can enhance the efficiency of domain calibration in large parameter spaces.

## 5.4 Generalization across Various Conditions

Madoc leverages static offline datasets to adjust the domain parameters, driving our curiosity toward its generalization capacity under varying dataset sizes and initial parameter ranges of the source domain. Here we choose all the above-mentioned tasks on the NeoRL benchmark and calculate the averaged normalized returns for comparison. As illustrated in Fig. 4(a), the algorithms access datasets of different magnitudes, $5 \times 10^4$ (small), $2 \times 10^5$ (medium), and $1 \times 10^6$ (large), to reflect a spectrum of data availability. The results reveal that the effectiveness of both the hybrid offline-and-online H2O algorithm and the purely offline MOREC algorithm rises with the expansion of the dataset size. This discovery suggests that their dependency on the size of offline data for improved performance. On the contrary, our method maintains stable and excellent performance, unfazed by the dataset size. Besides, for source domains with various initial parameter ranges, categorized as easy, medium, and hard (see App. D for more details), Madoc exhibits remarkable performances across all levels, particularly excelling in "hard" cases with the largest parameter search space, as shown in Fig. 4(b).

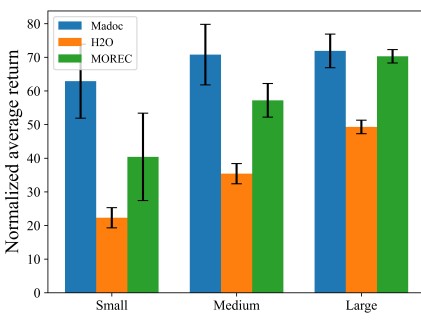
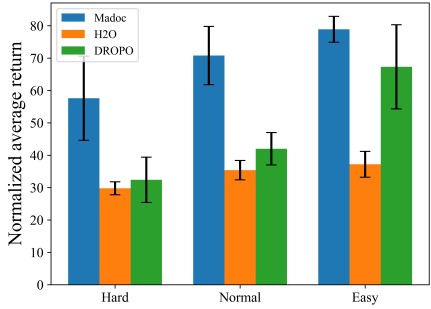

(a) Performance with different sized datasets

(b) Performance under different initial domain parameter ranges

Figure 4: The generalization ability of Madoc compared to the baselines under different conditions.

This underscores our algorithm's effectiveness in coping with challenging scenarios, affirming its robustness and adaptability in diverse conditions.

## 6 Conclusion and Discussion

In this paper, we introduce Madoc, a framework for closing the dynamics gap by calibrating the source domain with a handful of offline data via multi-agent reinforcement learning. Concretely, the target domain data serve as a guide for target transition dynamics, which is leveraged to train classifiers generating rewards and derive a bandit RL objective for domain calibration, To improve calibration efficiency with a large number of parameters, we further model it as a cooperative MARL problem and propose to group parameters with similar effects on dynamics. Experiments on popular control tasks demonstrate that our method can calibrate the source domain with sufficient accuracy, allowing the optimal trained policy to be transferred to the target domain without severe performance deterioration. One possible constraint of our method is that, when dealing with high-dimensional vision tasks [64], using a handful of offline data may not guarantee the accuracy and generalizability of the reward model. This challenge could be mitigated by deploying more expressively powerful tools like diffusion models [65], which is left for future work.

**Acknowledgements** This work is supported by the National Science Foundation of China (62276126, 62250069), the Natural Science Foundation of Jiangsu (BK20221442, BK20243039, and BK2024119), and the Fundamental Research Funds for the Central Universities (0221/14380022).

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

# Appendix

## A  Algorithm Description

The pseudo-code of Madoc is presented in Alg. 1, we utilize SAC [59] and DOP [49] as our backbone algorithms for domain calibration. From lines 1 to 9, we group the domain parameters by learning a VAE and employing k-means; lines 13 to 16 aim at training discriminators as the reward model and regularizing the running policy; the calibration actors and critics are updated from lines 17 to 19; finally, we train the running policy from scratch and deploy it to the target domain.

---

**Algorithm 1** Madoc

---

**Input:** A pre-collected dataset $\mathcal{D}$, an imperfect manipulable source domain $\mathcal{M}$
**Initialize:** Policy $\pi_\theta$, joint calibration actor $q_\phi$ and critic $v_\Phi$, a couple of binary discriminators $D_\psi$,
   VAE $f_\Psi$, prior distributions $p(\xi)$ and $p(z)$, replay buffer $\mathcal{B} = \emptyset$ for simulated rollouts, step for
   parameter grouping $T_{\text{group}}$, step for domain calibration $T_{\text{calibration}}$, learning rate $\eta$
 1: **for** step $t = 1, \cdots, T_{\text{group}}$ **do**
 2:    Behavior cloning:
 3:       $\pi_\theta \leftarrow \text{BC}(\pi_\theta, \mathcal{D})$
 4:    Collect simulated data:
 5:       $\mathcal{B} \leftarrow \mathcal{B} \bigcup \text{ROLLOUT}(\pi_\theta, q_\phi, \mathcal{M})$
 6:    Update VAE:
 7:       $f_\Psi \leftarrow f_\Psi - \eta \nabla_\Psi \mathcal{L}_\Psi$ according to Eq. 10
 8: **end for**
 9: Get $n$ groups by k-means, clear replay buffer $\mathcal{B} = \emptyset$, and reinitialize policy $\pi_\theta$
10: **for** step $t = T_{\text{group}} + 1, \cdots, T_{\text{group}} + T_{\text{calibration}}$ **do**
11:    Collect simulated data:
12:       $\mathcal{B} \leftarrow \mathcal{B} \bigcup \text{ROLLOUT}(\pi_\theta, q_\phi, \mathcal{M})$
13:    Update discriminators:
14:       $D_\psi \leftarrow D_\psi - \eta \nabla_\psi \mathcal{L}_\psi$ according to Eq. 6
15:    Behavior cloning:
16:       $\pi_\theta \leftarrow \text{BC}(\pi_\theta, \mathcal{D})$
17:    Update the joint calibration actor and critic:
18:       $v_\Phi \leftarrow v_\Phi - \eta \nabla_\Phi \mathcal{L}_\Phi$ according to Eq. 12
19:       $q_\phi \leftarrow q_\phi - \eta \nabla_\phi \mathcal{L}_\phi$ according to Eq. 13
20: **end for**
21: Train policy from scratch:
22:    $\pi_\theta \leftarrow \text{SAC}(\pi_\theta, q_\phi, \mathcal{M})$
23: **Output:** Policy $\pi_\theta$

---

## B  Extended Related Work

**Sample efficient RL.** A significant drawback of current reinforcement learning methods is their poor sample efficiency, leading to extensive environmental interactions [66]. This results in prohibitive costs in real-world applications [67] and hinders policy learning even in complex digital environments, like the full StarCraft game [68]. Multiple factors may restrict the sample efficiency, while several aspects can alleviate these limitations. Different optimization methods possess varying capabilities for exploring and exploiting data. Mainstream algorithms in reinforcement learning commonly rely on the gradient of the objective or surrogate objectives [69]. Still, there are also sample-based methods, a.k.a. derivative-free optimization, that offer their unique advantages. Typical algorithms like evolutionary algorithms [70] and Bayesian optimization [71] have been applied for reinforcement learning [72], showcasing better performance on some tasks. Nonetheless, these methods also encounter certain drawbacks, including slow convergence, difficulties in scaling, sensitivity to noise, and a lack of theoretical guarantee. From another perspective, model-based algorithms can be much more efficient as planning in the model is free of real-world samples in ideals [73]. Nevertheless, in high-dimensional environments, learning an accurate transition model using supervised learning is challenging. Employing manually constructed environments, i.e., simulators, is more efficient and practical, especially in the field of robotics [74]. However, it also encounters new problems due to

the sim2real dynamics gap, and simulators are generally non-differentiable, unlike neural models. Besides the aspects discussed above, the ability to transfer knowledge is also crucial for improving the sampling efficiency. Rather than starting from scratch for each task, humans continuously learn and build upon experiences from a variety of tasks. Many methods have proposed various types of knowledge transfer, such as policy transfer [75], representation transfer [76], and skill transfer [77]. However, these methods work effectively only in specific cases. A general approach to transfer reinforcement learning is yet to be developed.

**Offline RL.** In the offline RL setting, the agent no longer has the ability to interact with the environment [78]. Instead, the learning algorithm is access to a static offline dataset, collected previously by an unknown behavior policy from the target domain. The main obstacle to this data-driven learning paradigm is the distribution shift problem due to the discrepancy between the learned and behavior policies, leading to severe extrapolation error. Previous works tackle this problem by constraining the learned policies against the behavior policy [79] or penalizing the value function on out-of-distribution (OOD) actions [63], both of which require large enough datasets. Besides, offline model-based RL algorithms learn a dynamics model from offline data to enhance the efficiency of offline RL [80]. Benefiting from generating the synthetic data by the learned dynamics, model-based algorithms improve the coverage of the dataset. However, this depends on accurate dynamics models and still requires sufficient offline data. It is worth noting that in this work we use offline datasets to calibrate domain parameters, thereby avoiding the selection of OOD actions, and alleviating the requirements for dataset quality and quantity.

## C   Implementation Details

**Discriminators.** We train two discriminator networks $D_{\psi_{\mathrm{sas}}}(\cdot|s, a, s')$ and $D_{\psi_{\mathrm{sa}}}(\cdot|s, a)$ to classify whether the rollouts comes from the offline dataset or the source domain in our method. We follow the implementation in DARC [29], propagating gradients back through both discriminators as follows:

$$D_{\psi_{\mathrm{sa}}}(\cdot|s, a) = \mathrm{SoftMax}(f_{\mathrm{sa}}(s, a)),$$
$$D_{\psi_{\mathrm{sas}}}(\cdot|s, a, s') = \mathrm{SoftMax}(f_{\mathrm{sas}}(s, a, s') + f_{\mathrm{sa}}(s, a)),$$

where $f_{\mathrm{sa}}(s, a)$ and $f_{\mathrm{sas}}(s, a, s')$ represent the outputs of the two classifier networks, $\mathrm{SoftMax}(x_i) = \frac{\exp x_i}{\sum_{x_j \in X} \exp x_j}, X = \{\mathrm{target}, \mathrm{source}\}$. Both networks contain two hidden layers, while dropout layers and ReLU activations are used between layers. Before the final $\mathrm{SoftMax}$ layer, we employ the Tanh activation function to map the outputs into probabilities.

**Grouping VAE.** We train both the encoder and decoder with two hidden layers, both dropout layers and ReLU activation functions are used between layers. We employ MSE loss for state reconstruction and eliminate KL divergence, as our goal is to encode the agent identity rather than generating states. For each domain parameter, we encode its identity into a 16-dimensional latent space. Then the latent variable is concatenated with the current observation and action, and the combined data is fed into the decoder to predict the observation for the next time step. The latent variables of all parameters are projected to three dimensions using Principal Component Analysis (PCA) [81], after which they are grouped using the k-means algorithm [58]. To reduce randomness in the group results, we repeatedly extract schemes from the last phase of group training and determine the final grouping scheme by choosing the one that occurs most frequently.

**Calibration policy.** We use DOP [49] as the multi-agent decomposition method because it is a popular off-policy multi-agent policy gradient method. As claimed in the paper, individual critics $v_\Phi^{gi}$ are learned by backpropagating gradients from global TD updates, guiding the updates of individual calibration actors by a similar objective as in SAC [59]:

$$\mathcal{L}_\phi = \mathbb{E}_{\xi^{gi} \sim q_\phi^{gi}(\cdot)} \left[ \alpha \log q_\phi^{gi}(\xi^{gi}) - v_\Phi^{gi}(\xi^{gi}) + \lambda D_{\mathrm{KL}} \left( q_\phi^{gi}(\xi^{gi}) || p^{gi}(\xi^{gi}) \right) \right].$$

Here, we do not employ the double-Q function and the automatic tuning scheme of $\alpha$. Instead we update $\lambda_t = \min(10, \exp(\frac{t}{2 \times 10^5}))$ and prior $p^{gi}(\xi^{gi})$ as a target network of $q^{gi}(\xi^{gi})$ with smoothing coefficient $1 \times 10^{-3}$ to stabilize training. The prior parameter $p^{gi}(\xi^{gi})$ is distribution is initialized as a normal distribution $\mathcal{N}(0, 1)$ to enhance exploration at the early stages of training.

We scale the output of the calibration actors by using a Gaussian with Tanh squashing, e.g., we rescale the output of *gravity* to be within $[-30, 0]$ in our preliminary experiments. To further enhance the

Table 3: The comparisons of method complexities.

| Method | GPU Memory Cost | Contained Modules |
|--------|-----------------|-------------------|
| Madoc | 398MB | reward models, grouping VAE, calibration agents, and running agents |
| CQL | 286MB | running agents |
| MOREC | 1053MB | dynamics reward function, dynamics models, and running agents |

Table 4: The comparisons of computational costs.

| Method | Total Training Time | Average Training Time per Epoch (over $1000$ epochs) |
|--------|---------------------|-------------------------------------------------------|
| Madoc | 5 hours | 1.8s for grouping (200 epochs), 14s for calibration, 4s for SAC |
| CQL | 2 hours | 7s for policy training |
| MOREC | 6 hours | 5s for training dynamics reward function, 16s for policy training |

Table 5: The common hyper-parameters in Madoc.

| Attribute | Value |
|-----------|-------|
| Calibration actor learning rate | $3 \times 10^{-4}$ |
| Calibration critic learning rate | $3 \times 10^{-4}$ |
| The temperature coefficient of calibration actor | 0.05 |
| Hidden layers of the calibration critic network | $[64, 64]$ |
| Target network smoothing coefficient for calibration actor | $1 \times 10^{-3}$ |
| Target network update interval for calibration actor | 10 |
| Discriminator learning rate | $1 \times 10^{-3}$ |
| Hidden layers of the discriminator network | $[256, 256]$ |
| The dropout rate of the discriminator network | 0.2 |
| Running actor learning rate | $3 \times 10^{-4}$ |
| Running critic learning rate | $3 \times 10^{-4}$ |
| Hidden layers of the running actor network | $[256, 256]$ |
| Hidden layers of the running critic network | $[256, 256]$ |
| Target network smoothing coefficient for running actor | $5 \times 10^{-3}$ |
| Target network update interval for running actor | 1 |
| VAE learning rate | $3 \times 10^{-4}$ |
| Hidden layers of the VAE | $[256, 256]$ |
| Latent variable dimension of VAE | 16 |
| The dropout rate of the VAE | 0.5 |
| Batch size | 256 |
| Optimizer | Adam |
| Discount factor $\gamma$ | 0.99 |
| Buffer size | $2 \times 10^5$ |

exploration of domain parameters, we sample a series of domain parameters every 100 rollouts in the source domain. After calibration, we set the physics parameters with the means outputted by actors, and train SAC from scratch with default hyper-parameters.

Most experiments were conducted on a server outfitted with a 13th Gen Intel(R) Core(TM) i9-13900K CPU, 2 NVIDIA RTX A5000 GPUs, and 125GB of RAM, running Ubuntu 22.04. We also conduct a comparison between Madoc and various offline RL algorithms with regard to method complexities and computational costs on the hfctah-med-rep task of the D4RL benchmark, the results are presented in Tab. 3 and Tab. 4. Compared to the traditional pure offline RL algorithm CQL, Madoc incorporates additional modules that lead to increased GPU memory cost. Furthermore, it encompasses three training stages, which consequently require more computational cost. Nevertheless, given the significant performance improvement, these extra expenditures are deemed justifiable. We list the default hyper-parameter settings for Madoc in Tab. 5.

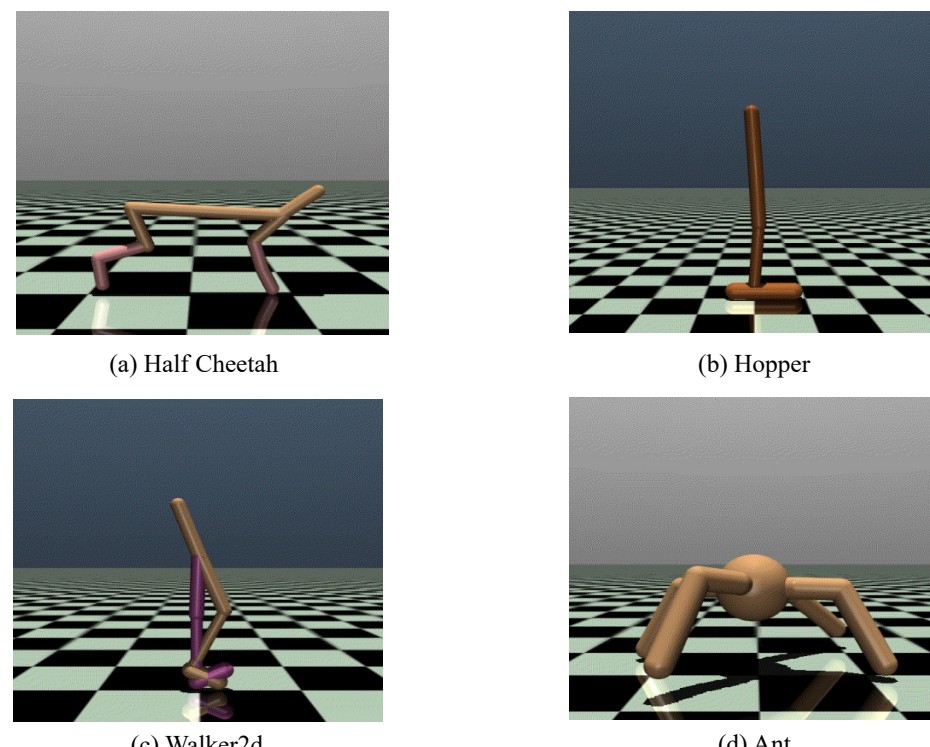

(a) Half Cheetah        (b) Hopper

(c) Walker2d        (d) Ant

Figure 5: An illustration of environments used in our experiments.

# D  Experiment Details

## D.1  Extended Environment Descriptions

We adopt the popular Gym-MuJoCo tasks [53] as our benchmarks used in the experiments. To investigate the performance of our method and baselines, we conduct experiments on four locomotion tasks shown in Fig. 5, including *HalfCheetah*, *Hopper*, *Walker2d*, *Ant*.

**HalfCheetah** is a 2D robotic simulation featuring 9 links and 8 joints, including two paws, designed to move forward rapidly by applying torque on 6 specific joints. The main objective of this robot centers on maximizing forward progression to earn positive rewards while minimizing backward movement to avoid penalties.

**Hopper** is designed to enhance the complexity with more state and control variables than traditional control settings, using a two-dimensional, one-legged hopper composed of a torso, thigh, leg, and foot. The overall aim is to propel the hopper forward by exerting torque on the three joints that link its four body sections.

**Walker2d** is a two-dimensional figure with two legs, including a torso, two thighs, two legs, and two feet, focusing on coordinated movement to advance. The objective is to utilize torque on six joints to synchronize the movement of all six body parts toward the desired direction.

**Ant** is a 3D model with a freely rotating torso and four two-linked legs, designed for coordinated forward movement. The aim is to maneuver the ant by applying torques on eight hinges to effectively control the movement of its nine parts toward the desired direction.

The number of groups $n$ for each task is 6, 4, 6, and 6, respectively. We verify in extended experiments (shown in Fig. 7(a)) that the performance results do not differ significantly as long as it is within a reasonable range. We report the initial ranges and ground truth value of each physics parameter in Tab. 6, Tab. 7, Tab. 8, and Tab. 9.

Table 6: The setting of *HalfCheetah* physics parameter at easy/normal/hard level.

| Physics Parameter | Initial Range (easy/normal/hard) | Ground Truth |
|---|---|---|
| *gravity* | $[-15, -5]/[-30, 0]/[-50, 0]$ | $-9.81$ |
| *body_mass_1* | $[6, 7]/[5, 7]/[4, 8]$ | 6.36 |
| *body_mass_2* | $[1, 2]/[0, 2]/[0, 3]$ | 1.54 |
| *body_mass_3* | $[1, 2]/[0, 2]/[0, 3]$ | 1.58 |
| *body_mass_4* | $[0.5, 1.5]/[0, 2]/[0, 3]$ | 1.07 |
| *body_mass_5* | $[1, 2]/[0, 2]/[0, 3]$ | 1.43 |
| *body_mass_6* | $[0.5, 1.5]/[0, 2]/[0, 3]$ | 1.18 |
| *body_mass_7* | $[0.5, 1.5]/[0, 2]/[0, 3]$ | 0.85 |
| *dof_damping_3* | $[4.5, 7.5]/[1.5, 7.5]/[0, 7.5]$ | 6 |
| *dof_damping_4* | $[3, 6]/[1.5, 7.5]/[0, 7.5]$ | 4.5 |
| *dof_damping_5* | $[1.5, 4.5]/[1.5, 7.5]/[0, 7.5]$ | 3 |
| *dof_damping_6* | $[3, 6]/[0, 6]/[0, 7.5]$ | 4.5 |
| *dof_damping_7* | $[1.5, 4.5]/[0, 6]/[0, 7.5]$ | 3 |
| *dof_damping_8* | $[0, 3]/[0, 6]/[0, 7.5]$ | 1.5 |

Table 7: The setting of *Hopper* physics parameter at easy/normal/hard level.

| Physics Parameter | Initial Range (easy/normal/hard) | Ground Truth |
|---|---|---|
| *gravity* | $[-15, -5]/[-30, 0]/[-50, 0]$ | $-9.81$ |
| *body_mass_1* | $[3, 4]/[3, 5]/[2, 6]$ | 3.53 |
| *body_mass_2* | $[3.5, 4.5]/[3, 5]/[2, 6]$ | 3.93 |
| *body_mass_3* | $[2, 3]/[2, 4]/[1, 5]$ | 2.71 |
| *body_mass_4* | $[4.5, 5.5]/[4, 6]/[3, 7]$ | 5.08 |
| *dof_damping_3, 4, 5* | $[0, 2]/[0, 3]/[0, 4]$ | 1 |

Table 8: The setting of *Walker2d* physics parameter at easy/normal/hard level.

| Physics Parameter | Initial Range (easy/normal/hard) | Ground Truth |
|---|---|---|
| *gravity* | $[-15, -5]/[-30, 0]/[-50, 0]$ | $-9.81$ |
| *body_mass_1* | $[3, 4]/[3, 5]/[2, 6]$ | 3.53 |
| *body_mass_2, 5* | $[3.5, 4.5]/[3, 5]/[2, 6]$ | 3.93 |
| *body_mass_3, 6* | $[2, 3]/[2, 4]/[1, 5]$ | 2.71 |
| *body_mass_4, 7* | $[2.5, 3.5]/[2, 4]/[1, 5]$ | 2.94 |
| *dof_damping_3, 4, 5, 6, 7, 8* | $[0, 0.2]/[0, 0.5]/[0, 1]$ | 0.1 |

Table 9: The setting of *Ant* physics parameter at normal level.

| Physics Parameter | Initial Range | Ground Truth |
|:---:|:---:|:---:|
| *gravity* | $[-30, 0]$ | $-9.81$ |
| *body_mass_*$1$ | $[0, 0.5]$ | $0.33$ |
| *body_mass_*$2, 3, 5, 6$ | $[0, 0.1]$ | $0.036$ |
| *body_mass_*$4, 7$ | $[0, 0.1]$ | $0.065$ |
| *dof_damping_*$6, 7, 8, 9$ | $[0, 3]$ | $1$ |

## D.2 Baselines

Here we introduce the baselines used in our experiments, including offline domain calibration, hybrid offline-and-online RL, and pure offline RL algorithms.

**DROPO** [17] adapts a distribution of dynamics parameters to match an offline dataset by employing a probabilistic distance measure, aimed at directly maximizing the likelihood of replicating real-world data within a simulation. Consequently, the simulator can be envisioned as a stochastic forward model, where the inherent randomness is attributed to variations in the scene's physical parameters.

**DROID** [16] harnesses human demonstrations to synchronize the simulator's trajectories with those observed in the real world, rather than relying on guesswork or exhaustive adjustments to establish the domain randomization (DR) range. This process helps in finding the most suitable range of parameters for the simulator, which can be formulated as a statistical model. Subsequently, this model can be sampled to inform the training process of RL agents.

**OTED** [51] is designed to autonomously learn a set of simulator parameters that align with a given offline dataset. Using the calibrated simulator, it proceeds to train a Reinforcement Learning (RL) agent using conventional online methods. An objective is formulated for the tuning of simulator parameters, aiming to minimize a divergence metric between the state-action distribution generated by the simulator and the provided target offline dataset.

**H2O** [62] presents a novel policy evaluation framework that is aware of dynamics, adaptively imposing penalties on Q-function training for simulated state-action pairs that exhibit significant dynamics discrepancies. At the same time, it permits learning from a predetermined dataset originating from the real world without direct interactions with it.

**CQL** [63] enhances the conventional Bellman error objective by adding an uncomplicated Q-value regularization term, which is easy to apply to most current deep Q-learning and actor-critic models. The objective of CQL is to overcome existing limitations by training a conservative Q-function, ensuring that the policy's expected value, as estimated by this Q-function, is a conservative estimate of its actual value, thus avoiding selecting OOD actions.

**MOREC** [52] acquires a dynamics reward function that can be generalized from offline data. This reward function is then utilized as a transition filter within any offline Model-Based Reinforcement Learning (MBRL) approach. During the transition generation process, the dynamics model produces a set of possible transitions, from which the one with the highest dynamics reward value is chosen for selection and used for policy update.

## E   Additional Experiment Results

### E.1   The Absolute Calibration Error

We report the corresponding mean absolute calibration error of experiments in Sec. 5.2. As shown in Tab. 10 and Tab. 11, there are five methods performing domain transfer by tuning simulator parameters. It is apparent that the evolutionary algorithm-based methods, DROPO and DROID, result in huge calibration errors, which suggests their limited effectiveness within the realm of high-dimensional parameter spaces. Additionally, Madoc-S outperforms OTED, highlighting the more rational design

Table 10: The mean absolute calibration error on D4RL benchmark. We **bold** the lowest mean.

| Task | DROPO | DROID | OTED | Madoc-S | Madoc |
|------|-------|-------|------|---------|-------|
| hfctah-med | 0.78±0.18 | 1.13±0.34 | 0.28±0.03 | 0.13±0.04 | **0.06±0.01** |
| hfctah-med-rep | 0.64±0.15 | 1.35±0.40 | 0.61±0.11 | 0.23±0.10 | **0.12±0.02** |
| hfctah-med-exp | 0.66±0.14 | 0.87±0.22 | 0.33±0.06 | **0.10±0.03** | 0.09±0.02 |
| hopper-med | 1.20±0.35 | 0.74±0.29 | 0.42±0.05 | **0.34±0.07** | 0.33±0.06 |
| hopper-med-rep | 1.27±0.38 | 1.21±0.33 | 0.48±0.07 | 0.53±0.07 | **0.39±0.05** |
| hopper-med-exp | 0.73±0.23 | 1.43±0.40 | 0.65±0.35 | 0.38±0.08 | **0.34±0.06** |
| walker-med | 0.55±0.16 | 0.45±0.09 | 0.30±0.05 | 0.26±0.05 | **0.18±0.02** |
| walker-med-rep | 1.16±0.20 | 1.13±0.34 | 0.38±0.07 | 0.33±0.05 | **0.25±0.04** |
| walker-med-exp | 0.64±0.12 | 0.48±0.13 | 0.29±0.04 | 0.21±0.04 | **0.19±0.03** |
| ant-med | 1.12±0.27 | 1.04±0.29 | 0.41±0.11 | 0.25±0.04 | **0.16±0.02** |
| ant-med-rep | 0.74±0.22 | 0.84±0.31 | 0.38±0.12 | 0.25±0.06 | **0.14±0.02** |
| ant-med-exp | 0.54±0.13 | 0.60±0.30 | 0.27±0.06 | 0.21±0.07 | **0.14±0.02** |

Table 11: The mean absolute calibration error on NeoRL benchmark. We **bold** the lowest mean.

| Task | DROPO | DROID | OTED | Madoc-S | Madoc |
|------|-------|-------|------|---------|-------|
| HalfCheetah-L | 0.55±0.14 | 0.40±0.10 | 0.46±0.07 | 0.43±0.27 | **0.13±0.04** |
| HalfCheetah-M | 0.92±0.37 | 0.73±0.23 | 0.34±0.04 | 0.24±0.03 | **0.20±0.05** |
| HalfCheetah-H | 0.50±0.17 | 0.73±0.22 | 0.47±0.05 | 0.32±0.06 | **0.21±0.05** |
| Hopper-L | 0.76±0.19 | 0.59±0.22 | 0.30±0.05 | **0.22±0.06** | 0.24±0.05 |
| Hopper-M | 0.99±0.21 | 1.06±0.29 | 0.42±0.07 | 0.34±0.08 | **0.26±0.04** |
| Hopper-H | 0.93±0.28 | 1.90±0.67 | 0.38±0.07 | 0.34±0.08 | **0.26±0.04** |
| Walker2d-L | 0.86±0.37 | 1.22±0.38 | 0.27±0.05 | 0.24±0.04 | **0.22±0.04** |
| Walker2d-M | 1.16±0.34 | 0.62±0.22 | 0.26±0.06 | 0.29±0.03 | **0.17±0.02** |
| Walker2d-H | 1.17±0.39 | 1.09±0.31 | 0.24±0.05 | 0.28±0.04 | **0.18±0.03** |

of our reward model. Madoc minimizes the mean absolute calibration errors in almost all tasks, which also serves as a basis for its efficient domain transfer.

### E.2 Ablation Studies

In order to further validate the effectiveness of the automatic grouping technique, as in Fig. 6, we implement a variant of Madoc, which is not equipped with the automatic grouping technique, i.e., there are $N$ agents each responsible for calibrating a single parameter. We set the same hyper-parameters, and train both policies on the *HalfCheetah* tasks of the NeoRL benchmark. We can find the policy performance of Madoc significantly decreases without the use of the automatic grouping technique, underscoring the importance of this component.

Besides, we additionally consider independent learning methods and different value decomposition methods. Independent learning [41] treats each agent as an independent individual, optimizing each policy with shared rewards without considering the joint policy, has gained traction again due to their surprising performance in some domains [82]. Therefore, we have implemented the Madoc-ISAC algorithm for domain calibration. VDN [44] is a classic value decomposition method representing the global value function as a simple sum of individual value functions, and we denote the corresponding version as Madoc-VDN. We train these policies with the same hyper-parameters on the *HalfCheetah* tasks of the D4RL benchmark. The results, shown in Tab. 12, demonstrate that Madoc-ISAC and Madoc-VDN perform worse than Madoc on all three datasets. All physics parameters in the source domain are interrelated and cooperative. Consequently, the independent learning method overlooks the policy changes of other calibration agents, which leads to non-stationary problems and a decline in performance. The gap between Madoc-VDN and Madoc is small, we speculate the reason is domain calibration is essentially a bandit RL problem, where there is no state space (therefore, we do not compare with more complex value decomposition algorithms [45] either, as their implementation

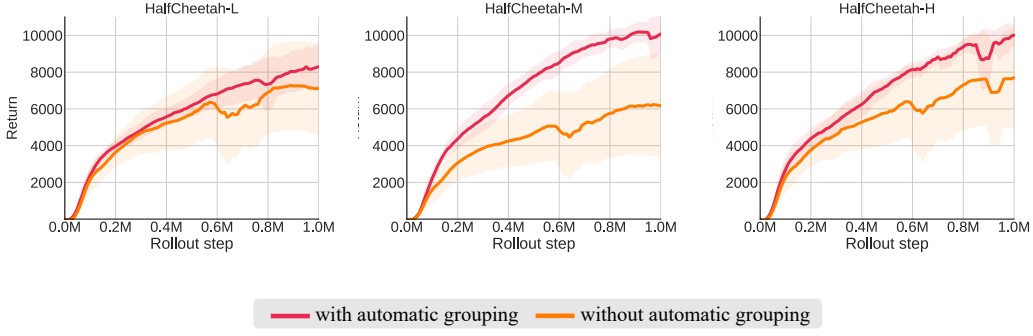

Figure 6: The average return of rollout steps in the source domain with or without the automatic grouping technique.

Table 12: Normalized average return of Madoc and its variants. The results are evaluated in the target domain and we **bold** the highest mean.

|  | **hfctah-med** | **hfctah-med-rep** | **hfctah-med-exp** |
|---|---|---|---|
| Madoc | **91.9**± **7.7** | **95.7**± **9.9** | **96.9**± **5.3** |
| Madoc-VDN | 89.9± 6.9 | 92.7± 7.1 | 88.2± 5.7 |
| Madoc-ISAC | 81.1±14.7 | 75.0±13.1 | 75.2±16.1 |

would be the same without the global state). The main purpose of value decomposition methods is to perform credit assignment and both methods can achieve this. This also reflects that our algorithm can be integrated with any existing MARL value decomposition methods.

### E.3  Stability of the Experimental Results

The performance results presented in Tab. 1 and Tab. 2 reveal that Madoc exhibits greater variance in certain task scenarios compared to offline RL algorithms. In this subsection, we explain the large variance and have designed corresponding adjustments to improve upon this instability.

Madoc has achieved a trade-off between high mean and low variance in return performance. On the one hand, Madoc requires online interaction with the source domain to search for the domain parameters that best match the offline dataset. Consequently, the random seed significantly influences exploration and exploitation, leading to a larger variance for Madoc. In contrast, pure offline algorithms like CQL and MOREC learn on a fixed dataset in a conservative manner and do not need to explore. Therefore, they are less influenced by the random seed and have smaller variance. On the other hand, algorithms with lower variance, namely H2O, DR+BC, CQL, and MOREC, obtain conservative policies by penalizing the value functions on OOD actions or directly constraining the policies against the behavior policies. As a result, their mean performances are also limited by the dataset. Our method has achieved a trade-off between high mean and low variance, attaining optimal performance compared to baselines in most scenarios.

Regarding the large variance problem of Madoc, we have made some improvements. Once domain calibration is completed, we no longer use pure SAC to train the policy on the source domain from scratch. Instead, we combine SAC with BC to impose appropriate constraints on the learned policy, referred to as Madoc+BC. The results are shown in the Tab. 13. We can observe that the mean performance of Madoc+BC decreased slightly but became more stable, confirming our approach.

### E.4  Sensitivity of Hyper-parameters

By utilizing the automatic grouping technique, Madoc clusters physical parameters into several groups; thus, we investigate the impact of the number of groups $n$ on algorithm performance in the hfctah-med-rep task. As illustrated in Fig. 7(a), the best performance is achieved when the number of groups is 6; however, the impact is not significant as long as the value is within an appropriate range.

Table 13: Normalized average return of Madoc and its variants. The results are evaluated in the target domain and we **bold** the highest mean.

|  | **hfctah-med** | **hfctah-med-rep** | **hfctah-med-exp** |
| --- | --- | --- | --- |
| Madoc | **91.9±7.7** | **95.7±9.9** | 96.9±5.3 |
| Madoc+BC | 88.9±4.6 | 90.1±4.8 | **97.2±3.3** |
| MOREC | 73.9±3.0 | 74.1±2.8 | 72.0±3.1 |

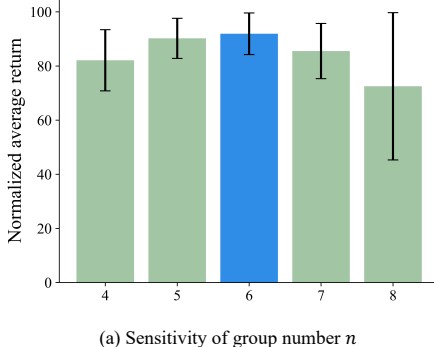

(a) Sensitivity of group number $n$

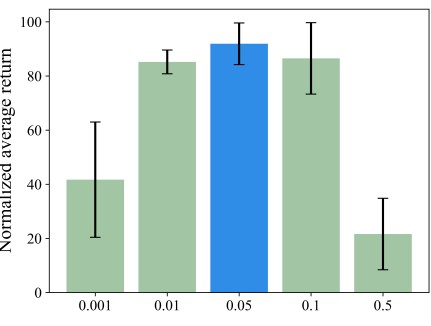

(b) Sensitivity of temperature coefficient $\alpha$

Figure 7: Sensitivity of hyper-parameters.

Therefore, we default to setting 6 as the number of groups for the three tasks *HalfCheetah*, *Walker2d*, *Ant*, and for the *Hopper* task, which has fewer domain parameters, we choose 4 as the number of groups. Additionally, for the $\alpha$ hyper-parameter that affects the entropy of the calibration policy during the domain calibration process, we also design experiments on this task to verify parameter sensitivity. Fig. 7(b) demonstrates that $\alpha = 0.05$ is the best choice, values that are too large or too small will both lead to reduced search efficiency, affecting the final policies' performance. We also set this hyper-parameter to the same value for all experiments, and it showcases stable results.

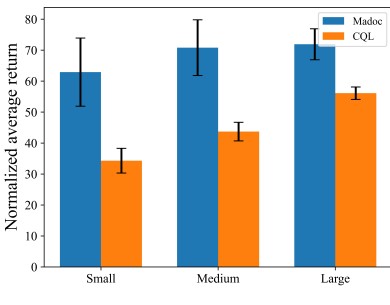

(a) Performance with different sized datasets

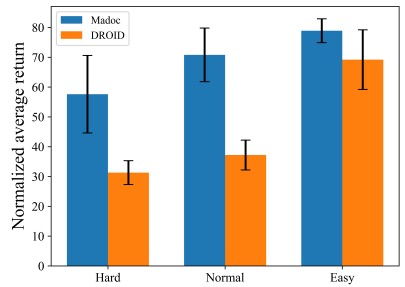

(b) Performance under different initial domain parameter ranges

Figure 8: More results about the generalization ability of Madoc compared to the baselines under different conditions.

### E.5   More Generalization Results

In Sec. 5.4, we have already verified the generalization capability of Madoc across different datasets and under different search spaces. Here, we list the comparative results with more baseline algorithms. As shown in Fig. 8, the pure offline RL algorithm CQL is inferior to Madoc across various datasets, highlighting the limitations of conservative algorithms; DROID employs an evolutionary algorithm for parameter optimization and can achieve performance comparable to Madoc in simple cases, but collapses in complex ones. Madoc, on the other hand, can stably handle different scenarios and achieve excellent performance, supporting its generalization stability.

