# OpenReview forum: "Multi-Agent Domain Calibration with a Handful of Offline Data"
_NeurIPS.cc/2024/Conference — NeurIPS 2024 poster_

### Official Review · Reviewer_EfDo · 2024-07-10

**Soundness:** 2
**Presentation:** 2
**Contribution:** 2
**Rating:** 6
**Confidence:** 3

**Summary:**

The paper addresses the challenge of performance degradation when RL policies trained in one domain are deployed in another with different dynamics. The proposed solution, Madoc (Multi-agent domain calibration), uses a small amount of offline data from the target domain to adjust the source domain's physics parameters. This approach improves upon existing domain calibration techniques by employing cooperative Multi-Agent Reinforcement Learning (MARL) to handle a large parameter space more efficiently.

**Strengths:**

- Strong empirical performance

**Weaknesses:**

- The paper is so hard to follow.
  - In 4.1, we start with KL minimization objective. It is difficult to understand how it is optimized, as the paper lacks sufficient explanation and some derivations are incorrect. Eq (4) seem to be completely different formula compared to eq (3). regarding q_\phi, in eq (4), the only relevant term is the third term, which makes no sense.
  - Section 4.2. refers to "calibaration critic" and some terms that are not defined.
  - especially the latter part of 4.2, explaining about the parameter grouping, is mostly not understandable. what does "the identity i of each parameter" mean? How can $i$ itself determine s' given s,a,\zeta^i without knowing \zeta^-i?

minor comments:
- in figure 2 legend, blue dots are said to be MA, whereas the text on left says blue dots are single-agent method.

**Questions:**

- Regarding the objective at the start, eq. (1), why do we need to match action distribution as well? Usually if the transition is varying, the objective is to match the state distributions. If the transitions are different, the actions to make the similar state distributions will be different, and it seems natural to not match the action distributions.
- Why do we need the multi-agent method? in section 4.2. it is motivated in a way that shared critic struggles to optimize policy in a huge action space. Does it mean that it is beneficial to use multi-agent algorithm for any RL environments for better optimization of policy? As far as I know, multi-agent algorithms are harder to optimize to the optimal policy (often converge to sub-optimal), and I am curious why this case is different.
- Why can't we apply algorithms like "imitation learning from observation" for this problem setting after concatenating the domain parameter to the action space? How is this paper different from such algorithms?

**Limitations:**

The authors adequately addressed the limitations.

---

> ### Author Rebuttal · Authors · 2024-08-07
>
> ### Q1: The objective in Sec. 4.1.
> **Eq. 4 is obtained through reasonable approximations.**
>
> The explanations for each derivation step are as follows:
> - Eq. 2 is derived from Eq. 1 based on the definitions of trajectory distribution and KL divergence, transforming the objective into the expectation under the parameter distribution and trajectory distribution of the source domain.
> - Eq. 3 further decomposes Eq. 2. Since $\log \frac{q_{\phi}(\xi)}{p(\xi)}$ is independent of the trajectory distribution $d_{\pi,\mathcal{M}_{\xi}}$, the expectation can be decomposed into two terms.
> - Eq. 4 approximates Eq. 3. As the parameter distribution and trajectory distribution used to calculate the expectation are difficult to compute, we use **Monte Carlo sampling** on the source domain to approximate the expected results. To enhance sampling efficiency, we store the samples in the replay buffer, **converting the trajectory-based objectives into transition-based objectives**, following the off-policy RL paradigm.
>
> We will add the detailed explanations in the revised manuscript.
> ### Q2: Some terms are not defined in Sec. 4.2.
> The calibration critic and the calibration policy (actor) together constitute a calibration agent. The calibration critic is responsible for evaluating the accuracy of the parameters output by the calibration actor. The absolute calibration error represents the absolute difference between the parameters output by the calibration actor and the target parameters.
> ### Q3: The latter part of Sec. 4.2 is not understandable.
> "The identity $i$ of each parameter" represents a specific identification symbol for each parameter. Specifically, in the experiments, these identities refer to a set of one-hot encodings used to distinguish different parameters.
> Besides, some key contents were missed: the identity can determine $s'$  given $s, a, \xi$ by **keeping the other parameters $\xi^{-i}$ fixed**, similar to the approach in COMA [1]. By only changing $\xi_i$, the effect of this parameter on the dynamics can be reflected, thereby facilitating parameter grouping .
> We apologize for the confusion and will add explanations in the revised manuscript for better understanding.
> ### Q4: Why do we need to match action distribution?
> **We need to match the action distribution to make the dynamics of the source domain closer to that of the target domain.**
>
> Eq. 1 is used to match the trajectory distribution (state distribution and action distribution). The underlying motivation is to minimize the transition dynamics gap. If only the state distribution is matched, multiple combined solutions of the policy and the transition function can exist. Therefore, to obtain the target transition function, we need to match the action distribution as well. To verify our idea, we conducted relevant experiments: for Eq. 1, we no longer matched the action distribution, denoted as Madoc_st. The experimental results, shown below, indicate performance degradation in all three scenarios.
> | Task           | Madoc    | Madoc_st  |
> | -------------- | -------- | --------- |
> | hfctah-med     | 91.9±7.7 | 78.2±8.9  |
> | hfctah-med-rep | 95.7±9.9 | 85.2±12.4 |
> | hfctah-med-exp | 96.9±5.3 | 88.1±10.1 |
> ### Q5: Why do we need the multi-agent method?
> **We use the multi-agent method for better optimization efficiency.**
>
> The search space of domain parameters grows exponentially with the number of parameters, hindering the optimization of both single-agent methods and previous multi-agent methods. Fortunately, with the advent of linear value factorization algorithms like VDN [2], researchers have improved scalability by decomposing the joint action space theoretically and practically [3], resulting in a much lower action space for each agent. Consequently, Madoc has higher optimization efficiency and can calibrate parameters more accurately.
> Certainly, the multi-agent method is not applicable to all RL environments. When there is only one agent in the environment with a low action dimension, a single-agent RL method might be more straightforward, providing satisfactory policy optimization without the added complexity.
> ### Q6: How about methods like "imitation learning from observation"?
> **Madoc is different from and superior to these methods.**
>
> Imitation from observation [4] methods (denoted as IfO) can be formalized as learning a discriminator $D(s, s')$ and a policy $\pi(a,\xi|s)$ simultaneously, while Madoc involves training a discriminator $D(s, a, s')$, a calibration policy $q(\xi)$ and a running policy $\pi(a|s)$. There are mainly two differences:
> - Madoc decouples tuning the domain parameters and the policy, making it more accurate and efficient in reducing the dynamic gap.
> - IfO requires matching the state distribution of the expert trajectory, which imposes additional quality requirements on the dataset [5]. However, Madoc only requires datasets of arbitrary quality to match the transition dynamics.
>
> We also tested the performance of IfO. The first table below reports the mean return during the test stage.
> | Task           | Madoc    | IfO       |
> | -------------- | -------- | --------- |
> | hfctah-med     | 91.9±7.7 | 43.2±28.9 |
> | hfctah-med-rep | 95.7±9.9 | 55.2±19.4 |
> | hfctah-med-exp | 96.9±5.3 | 42.1±22.1 |
>
> The second table below compares the mean absolute calibration error.
> | Task           | Madoc     | IfO       |
> | -------------- | --------- | --------- |
> | hfctah-med     | 0.06±0.01 | 0.98±0.12 |
> | hfctah-med-rep | 0.12±0.02 | 1.02±0.30 |
> | hfctah-med-exp | 0.09±0.02 | 0.79±0.08 |
>
> Reference:
>
> [1] Counterfactual Multi-Agent Policy Gradients. AAAI, 2018.
>
> [2] Value-Decomposition Networks for Cooperative Multi-Agent Learning Based on Team Reward. AAMAS, 2018.
>
> [3] Towards Understanding Cooperative Multi-Agent Q-Learning with Value Factorization. NeurIPS, 2021.
>
> [4] Recent Advances in Imitation Learning from Observation. IJCAI, 2019.
>
> [5] Robust Learning from Observation with Model Misspecification. AAMAS, 2022.

---

> > ### Comment · Reviewer_EfDo · 2024-08-10
> > **About author responses**
> >
> > Thanks for the detailed response. These responses address some of my concerns. However:
> >
> > 1. My question on (4) was that it no longer includes $q_\phi$ except the last term. optimizing it results in $q_\phi=p$, and I am sure that it is not what the paper is aiming to do.
> >
> > 2. I am still not convinced that multi-agent approach is mandatory in small-sized experiments of Mujoco locomotion tasks. I believe that the optimization challenge addressed with multi-agent approach can be also handled with complex neural architectures as Transformers/Diffusion models. In this sense, considering the difficulty in optimizing multi-agent algorithms, there won't be much interested researchers willing to explore similar directions.
> >
> > Due to these reasons, for now, I decide to keep the score.

---

> > > ### Comment · Area_Chair_crr9 · 2024-08-10
> > > **RE multi-agent approach**
> > >
> > > Thanks for weighing in!
> > >
> > > Can you clarify your comment that you are "not convinced that multi-agent approach is mandatory"?
> > > * Do you view this as a (e.g. central) claim of the work?
> > > * Is this related to some weakness or limitation of the experimental results?  What experiments would you want to see in order to be convinced?
> > > * Are there any particular works you can mention that use "complex neural architectures as Transformers/Diffusion models" to solve these or similar problems?  Or can you expand on how you envision this working?

---

> > > > ### Comment · Reviewer_EfDo · 2024-08-11
> > > >
> > > > Dear AC,
> > > >
> > > > My concerns are resolved by the authors, especially from the point that the proposed method only requires particularly small amount of data, and change in the equations. - and I raised the score.
> > > >
> > > > Thanks!

---

> > > ### Author Response · Authors · 2024-08-11
> > >
> > > Thank you once again for your thorough review and enthusiastic discussion.
> > > Regarding your first concern, we have decided to revise Eq. (4) from
> > > $$\approx-\mathbb{E}\_{(s,a)\sim\mathcal{B}}\left[\log\frac{\mu(a|s)}{\pi(a|s)}\right] - \mathbb{E}\_{(s,a,s',\xi)\sim\mathcal{B}}\log\left[\frac{T(s'|s,a,\xi^*)}{T(s'|s,a,\xi)}\right]+D\_{\mathrm{KL}}(q\_{\phi}(\xi)||p(\xi)),$$
> > > to
> > > $$\approx-\mathbb{E}_{a\sim\pi(\cdot|s)}\left[\log\frac{\mu(a|s)}{\pi(a|s)}\right]-\mathbb{E}\_{\substack{\xi\sim q\_\phi(\cdot) \\\\ (s,a,s')\sim d\_{\pi,\mathcal{M}\_{\xi}}(\cdot)}}\log\left[\frac{T(s'|s,a,\xi^*)}{T(s'|s,a,\xi)}\right]+D\_{\mathrm{KL}}(q\_{\phi}(\xi)||p(\xi)),$$
> > >
> > > where both the second and third terms optimize the parameter distribution $q_\phi$. While the underlying approach remains the same, the revised expression is now clearer. Our initial intention was to emphasize the off-policy training paradigm by introducing the replay buffer $\mathcal{B}$ in the equation, but this led to some confusion. We will update this section in the revised version and add an explanation to address your concern.
> > >
> > > Furthermore, addressing the concern about the necessity of a multi-agent approach, it's important to emphasize that the motivation behind using multi-agent methods is to mitigate optimization challenges posed by the large number of parameters. By employing the widely used value decomposition approach, the multi-agent method reduces the search space for each individual agent while preserving cooperation between different parameters. Additionally, we have chosen MARL for the following reasons:
> > >
> > > - With the advancement of deep learning and the development of techniques such as value decomposition in MARL [1, 2], research in multi-agent reinforcement learning (MARL) has made significant strides, overcoming numerous challenges and achieving notable progress across various fields. Consequently, the optimization challenges in multi-agent algorithms have been substantially alleviated in recent years.
> > >
> > > - Transformers and Diffusion models have recently shown exceptional results across various domains of deep learning. In reinforcement learning, approaches like TT [3], Diffuser [4], and DD [5] have also achieved impressive outcomes using these models. However, their success heavily relies on large datasets—TT, Diffuser, and DD required 1e6, 1e6, and 2.5e6 transitions, respectively, for training on continuous control tasks, while our method uses only 5e4 transitions. Consequently, their performance may not be as strong in our small-sample setting. Nonetheless, in future work, we plan to explore integrating Transformers and Diffusion models into our framework to enhance the expressive power of the reward model, particularly in scenarios involving high-dimensional image and text inputs.
> > >
> > > - Multi-agent methods can solve problems that single-agent approaches cannot. For example, MA-DAC [6] models dynamic algorithm configuration problems as multi-agent systems and addresses them using cooperative MARL algorithms, thereby improving optimization efficiency. Similarly, MA2ML [7] effectively tackles optimization learning challenges in the connection of modules in automated machines through MARL. The study in [8] introduces a generic game-theoretic credit assignment framework to enable decentralized execution in continuous control. MARLYC [9] presents a novel method called MARL yaw control, which optimizes the yaw of each turbine, ultimately enhancing the total power generation of the farm. Additionally, the work in [10] models image data augmentation as a multi-agent problem, offering a more fine-grained automatic data augmentation approach by dividing an image into multiple grids and determining a jointly optimal enhancement policy. These successful applications highlight the effectiveness of modeling problems as multi-agent systems.
> > >
> > > Thank you for your valuable feedback. We will continue to refine our framework to enhance domain transfer, guided by the reviewers' suggestions, and further advance the practical application of MARL.
> > >
> > > Reference:
> > >
> > > [1] Multi-agent deep reinforcement learning: a survey. Artificial Intelligence Review, 2022.
> > >
> > > [2] A survey of progress on cooperative multi-agent reinforcement learning in open environment. arXiv, 2023.
> > >
> > > [3] Offline reinforcement learning as one big sequence modeling problem. NeurIPS, 2021.
> > >
> > > [4] Planning with diffusion for flexible behavior synthesis. ICML, 2022.
> > >
> > > [5] Is conditional generative modeling all you need for decision making? ICLR, 2023.
> > >
> > > [6] Multi-agent dynamic algorithm configuration. NeurIPS, 2022.
> > >
> > > [7] Multi-agent automated machine learning. CVPR, 2023.
> > >
> > > [8] Multiagent model-based credit assignment for continuous control. AAMAS, 2022.
> > >
> > > [9] Marlyc: Multi-agent reinforcement learning yaw control. Renewable Energy, 2023.
> > >
> > > [10] Local patch autoaugment with multi-agent collaboration. IEEE Transactions on Multimedia, 2023.

---

> > > > ### Comment · Reviewer_EfDo · 2024-08-11
> > > >
> > > > Thanks for addressing my concerns. I changed my mind to raise the score.

---

### Official Review · Reviewer_dRio · 2024-07-13

**Soundness:** 3
**Presentation:** 2
**Contribution:** 2
**Rating:** 5
**Confidence:** 3

**Summary:**

This paper introduces Madoc, a domain transfer method that calibrates a source domain using small amount of offline data from the target domain via multi-agent reinforcement learning (MARL). More concretely, MARL is used to tune physics parameters governing dynamics in the source domain to more closely match those in the target domain. Empirically, Madoc enables transfer in D4RL and NeoRL benchmark tasks.

**Strengths:**

2. The domain transfer problem is clear and the method is well-motivated.
1. The empirical analysis consider many relevant baselines and considers appropriate benchmark tasks.

**Weaknesses:**

I lean to reject primarlily because the paper's empirical analysis does not seem to support the two core claims on the paper -- namely that Madoc reduce the difference in dynamics between source and target domains and outperforms existing methods.

1. **Weak empirical results.** In most if not all tasks, the confidence regions for Madoc overlap with those for other baselines.
2. **It unclear if Madoc is indeed reducing the dynamics gap between source and target domains, since this gap is not evaluated in experiments.** Additional experiments should be included showing that Madoc indeed reduces the dynamics gap. First instance, one could report the difference between source and target domain physics parameters and/or the KL divergence between source and target domains. As it stands, it looks like Madoc offers little to no improvement over baselines, which leads me to believe that Madoc might not be reducing the gap by much.

**Questions:**

1. I'm a bit confused by Figure 2. Could the authors provide more detail on what is being plotted here and why it is being plotted? In particular, what exactly are at the plotted metrics and what are they intuitively related?

**Limitations:**

Limitation are noted in the conclusion.

---

> ### Author Rebuttal · Authors · 2024-08-07
>
> ### Q1: Weak empirical results.
> **Our method Madoc has achieved a trade-off between high mean and low variance in return performance.**
> - On the one hand, Madoc requires online interaction with the source domain to search for the domain parameters that best match the offline dataset. Consequently, the random seed significantly influences exploration and exploitation, leading to larger variance for Madoc. In contrast, pure offline algorithms like CQL and MOREC learn on a fixed dataset in a conservative manner and do not need to explore. Therefore, they are less influenced by the random seed and have smaller variance.
> - On the other hand, algorithms with lower variance, namely H2O, DR+BC, CQL, and MOREC, obtain conservative policies by penalizing the value functions on OOD actions or directly constraining the policies against the behavior policies. As a result, their mean performances are also limited by the dataset. Our method has achieved a trade-off between high mean and low variance, attaining optimal performance compared to baselines in most scenarios.
>
> Regarding the large variance problem of Madoc, we have made some improvements. Once domain calibration is completed, we no longer use pure SAC to train the policy on the source domain from scratch. Instead, we combine SAC with BC to impose appropriate constraints on the learned policy, referred to as Madoc+online_bc. The results are shown in the table below. We can observe that the mean performance of Madoc+online_bc decreased slightly but became more stable, confirming our approach.
>
> | Task           | MOREC    | Madoc    | Madoc+online_bc |
> | -------------- | -------- | -------- | --------------- |
> | hfctah-med     | 73.9±3.0 | 91.9±7.7 | 88.9±4.6        |
> | hfctah-med-rep | 74.1±2.8 | 95.7±9.9 | 90.1±4.8        |
> | hfctah-med-exp | 72.0±3.1 | 96.9±5.3 | 97.2±3.3        |
>
> The concern you mentioned is of crucial significance, and we will incorporate its discussion in the experimental section of the revised manuscript.
>
> ### Q2: It's unclear if Madoc is indeed reducing the dynamics gap between the source and target domains.
> **Our method Madoc indeed reduces the dynamics gap between the source and target domains.**
> Some experimental results are as follows:
>
> | Task           | DROPO     | OTED      | Madoc     |
> | -------------- | --------- | --------- | --------- |
> | hfctah-med     | 0.78±0.18 | 0.28±0.33 | 0.06±0.01 |
> | hfctah-med-rep | 0.64±0.15 | 0.61±0.11 | 0.12±0.02 |
> | hfctah-med-exp | 0.66±0.14 | 0.33±0.06 | 0.09±0.02 |
>
> As shown in the table, we used "mean absolute calibration error" (i.e., the absolute difference between the physics parameters of the source and target domains) to measure the dynamics gaps of different algorithms. It can be observed that Madoc minimized the mean absolute calibration errors in all three tasks. The complete results are presented in Appendix E.1. We sincerely apologize for the confusion caused by the lack of appendix guidance in the manuscript and will add this guidance in the revised version.
>
> ### Q3: Could the authors provide more details about Fig. 2?
> **Fig. 2 demonstrates that, compared to the single-agent method, the multi-agent method can more effectively evaluate the accuracy of each physics parameter output by the calibration policy.**
> The calibration agent consists of a calibration critic that evaluates the accuracy of the physics parameters and a calibration actor (policy) that adjusts the output physics parameters. The plotted metrics are the values output by the calibration critic for a specific physics parameter (i.e., the gravity coefficient) under different parameter conditions. When the parameters output by the calibration actor are closer to the target parameters (indicating a smaller absolute calibration error), the evaluation value output by a "good" critic should be higher. The results in Fig. 2 show that our multi-agent method can train such a good critic, whereas the single-agent method fails to correctly evaluate the accuracy of the physics parameters due to the lack of reasonable reward allocation and the large search space.
> We have corrected the legend of Fig. 2 **in the left half of the PDF of the "global" response** and will provide more detailed explanations about it in the revised manuscript.

---

> > ### Comment · Reviewer_dRio · 2024-08-10
> >
> > Thank you for the response!
> >
> > * Ah, I see, the paper *does* compare the source/target calibration gap in Appendix E. I must've missed this; I don't think there's a direct reference to Appendix E in the main body. I agree, Madoc is indeed achieving a lower calibration gap. I'm raising my score now that this has been clarified. As a side note, I suggest adding a vertical bar separating Madoc-S and Madoc from the other baselines. When I was comparing numbers, I found myself accidentally comparing Madoc with Madoc-S rather than the baseline methods.
> > * Fig 2 is now clear to me. If the discussion in your rebuttal is incorporated into the main paper, readers like myself should have a much easier time understanding the punchline here. I think my confusion primarily arose because the legend was incorrectly labeled in the original submission.
> > * I'll make another comment about the performance (return) of Madoc vs other baselines later today once I have some time to carefully look over the results again. I'll respond as soon as I can to ensure we can discuss if needed!

---

> > > ### Author Response · Authors · 2024-08-12
> > >
> > > Thank you for your thoughtful feedback.
> > >
> > > We appreciate your suggestion about the vertical bar and will certainly consider it for the final revision. If you have any additional comments or concerns regarding the performance (return) of Madoc versus other baselines, we'd be happy to discuss them.

---

> > > > ### Comment · Reviewer_dRio · 2024-08-12
> > > >
> > > > Regarding returns achieved by Madoc vs baselines: Experiments show that Madoc is reducing the calibration gap, though I maintain that Madoc does not offer a significant performance gain in many tasks, and will keep my (updated) score. Given that Madoc is calibrates well, it would be worth better understanding why the gains are slim in the chosen tasks -- perhaps there are benchmarks that would better highlight the benefits of Madoc. Is there a simple toy task the authors could create where accurate calibration is necessary to achieve high performance? Such an example would be instructive.

---

> > > > > ### Author Response · Authors · 2024-08-13
> > > > >
> > > > > Thank you once again for your thorough review and engaging discussion. We would like to explain why accurate calibration is essential for achieving high performance from the following two perspectives:
> > > > >
> > > > > - **Performance Improvements with Accurate Calibration**: When the domain parameter space is large, Madoc significantly reduces the mean absolute calibration error to the lowest level on both benchmarks, leading to performance improvements of at least 28% and 23% over other domain calibration methods (DROPO, DROID, OTED, Madoc-S), respectively. This substantial boost in performance is attributed to accurate calibration, as all methods use the SAC algorithm after calibration is completed.
> > > > >
> > > > > - **Explaining the "Slim Gains"**: We observed that DROPO and DROID, despite their extremely poor parameter calibration, still manage to achieve mediocre performance. This can be explained by a "Floor Effect" [1], where, after reaching a certain threshold of poor calibration, further increases in calibration error do not result in significant performance declines. To validate this, we designed a toy task using the HalfCheetah environment, where only the gravity parameter requires calibration, with -9.81 as the target value. We trained policies on source domains with gravity coefficients of -10, -11, -12, -13, and -14, and then deployed them to the target domain. The performance changes are shown in the table below (tested with 4 seeds due to time constraints).
> > > > >
> > > > > | Gravity | Return    | $\Delta$ |
> > > > > | ------- | --------- | -------- |
> > > > > | -10     | 93.5±8.9  | 6.5      |
> > > > > | -11     | 68.2±17.6 | 25.3     |
> > > > > | -12     | 57.4±25.2 | 10.8     |
> > > > > | -13     | 47.1±17.6 | 10.3     |
> > > > > | -14     | 44.3±10.5 | 2.8      |
> > > > >
> > > > > In the table, $\Delta$ represents the change in performance between successive gravity coefficients. Mathematically, for a given gravity coefficient $g_i$ and its corresponding return $R(g_i)$, $\Delta$ is defined as:
> > > > >
> > > > > $
> > > > > \Delta = |R(g_{i-1}) - R(g_i)|
> > > > > $
> > > > >
> > > > > where $g_{i-1}$ and $g_i$ are consecutive gravity coefficients, with $g_{i-1}$ being closer to the target value of -9.81 than $g_i$. This metric illustrates how sensitive the performance is to changes in the gravity coefficient. We observed that as the gravity coefficient deviates further from the target value—from -10 to -13—the performance change $\Delta$ is substantial. However, as the gravity coefficient decreases further to -14, the performance change becomes less pronounced. This helps explain why "the gains are slim."
> > > > >
> > > > > **Reference:**
> > > > >
> > > > > [1] Floor and ceiling effects in the OHS: an analysis of the NHS PROMs data set. BMJ open, 2015.

---

### Official Review · Reviewer_Cdsr · 2024-08-02

**Soundness:** 3
**Presentation:** 3
**Contribution:** 3
**Rating:** 6
**Confidence:** 3

**Summary:**

The authors, introduce Madoc, a novel framework for domain calibration. By leveraging offline data from the target domain, it dynamically adjusts physics parameters, enabling direct policy deployment. To tackle the challenge posed by a large domain parameter space, the authors propose modeling domain calibration as a cooperative Multi-Agent Reinforcement Learning (MARL) problem. Experimental results demonstrate that Madoc surpasses existing techniques across most tasks in the D4RL and NeoRL benchmarks.

**Strengths:**

The paper addresses the critical research topic of domain calibration, which plays a pivotal role in enabling effective domain transfer of reinforcement learning (RL) methods. The authors present a well-written and organized paper, meticulously explaining their ideas and methodology. The motivation behind their approach is clearly articulated, and the experimental section thoughtfully addresses key questions. The experimental results are particularly compelling, demonstrating significant improvements over baselines on two standard benchmarks (D4RL and NeoRL). One of the paper’s novel contributions lies in its use of Cooperative Multi-Agent Reinforcement Learning (MARL) for adjusting source domain parameters. The authors delve into the specifics of their method, conducting ablation studies to highlight the impact of different components. Overall, I find this paper both interesting and potentially useful for the community.

**Weaknesses:**

*Major comments:*

- Method Complexity and Comparisons with Offline RL:

I noticed that the method appears more complex and computationally intensive compared to offline reinforcement learning (RL) methods like MOREC and CQL. This complexity might impact the fairness of the comparisons in Tables 1 and 2.
Adding discussion on the trade-off between complexity and performance in specific scenarios or domains could improve the paper.

- Surprising Results in Table 2:

I found it interesting that MOREC performs significantly poorly in HalfCheetah-M and Walker2d-H, which differs from the original paper results. It’s possible that differences in experimental settings, hyperparameters, or implementation details contribute to this discrepancy. Accurate baseline implementation is crucial for fair comparisons.

- Smallest Target Task Dataset Size:

The paper’s title suggests tackling domain transfer with handful of handful of offline data. The method seems to clearly perform well in small dataset regims. However, the smallest target task dataset still contains 5e4 samples. It’s reasonable to expect results with much smaller data sizes to better support the claim about domain transfer in sensitive contexts.

*Minor comments:*

In Section 4.2, the first paragraph refers to multi-agent (MA) methods depicted by red dots. However, the legend in Figure 2 uses blue for MA (presumably short for Multi-agent).

**Questions:**

- What is the computation cost of training your method compared to other Offline RL methods? Please add a discussion.

- The reward model seems to be a crucial component of this method. How good is the reward model? Is there a way to quantify its accuracy?

- Recently, in MARL community, independent learning methods have gained traction again due to their suprising performance in some domains [1] due to their robustness to large and complex tasks. How do you think independent SAC agents wll perform in this problem?

[1] de Witt et al, "Is Independent Learning All You Need in the StarCraft Multi-Agent Challenge?"

**Limitations:**

Please refer to the weaknesses section.

---

> ### Author Rebuttal · Authors · 2024-08-07
>
> ### Q1: Method Complexity and Comparisons with Offline RL.
> **Madoc adds more modules compared to CQL, but the model complexity and GPU memory cost remain acceptable.**
>
> We conducted experiments on the hfctah-med-rep task of the D4RL benchmark to evaluate the model complexity of different methods. The results are shown in the table below, with all networks in the form of MLP. Our method achieves a balance between model complexity and performance.
> | Method | GPU Memory Cost and Hidden Layers                                                                                                       |
> | ------ | --------------------------------------------------------------------------------------------------------------------------------------- |
> | Madoc  | 398MB: 2 * [256, 256] for reward models, 2 * [256, 256] for VAE, 6 * [64, 64] for calibration agents, 2 * [256, 256] for running agents |
> | CQL    | 286MB: 2 * [256, 256] for running agents                                                                                                |
> | MOREC  | 1053MB: [128, 256, 128] for dynamics reward function, 40 * [200, 200, 200, 200] for dynamics models, 2 * [256, 256] for running agents  |
> ### Q2: Surprising Results in Table 2.
> **The performance of MOREC is not as good as in the original paper because we used a smaller amount of offline data.**
> MOREC is sensitive to the quantity of offline data as it initially needs to train a generalizable dynamics reward function from offline data and accordingly learn dynamics models for generating transitions. When the quantity of offline data is insufficient, both the dynamics reward function and the dynamics models tend to underfit. As depicted in Fig. 4(a), when using the same large dataset as MOREC, the performance gap is small. However, as the amount of offline data decreases, the gap becomes increasingly larger. We will provide an explanation of it in the revised manuscript.
> ### Q3: Smallest Target Task Dataset Size.
> **Madoc utilized a sufficiently small amount of offline data.**
>
> We present the smallest target task dataset size of different methods below:
> | Method     | Size |
> | ---------- | ---- |
> | Madoc      | 5e4  |
> | SSR [1]    | 2e5  |
> | CQL, MOREC | 1e6  |
>
> SSR [1] proposes a data-efficient pipeline for offline RL with limited data and still used 2e5 training samples. This is because the continuous control tasks possess larger state and action spaces compared to discrete RL environments. Thus, the smallest target task dataset of Madoc, which contains only 5e4 samples, can be regarded as using a handful of offline data.
> ### Q4: What is the computation cost of training your method compared to other Offline RL methods?
> **Madoc incurs higher computational costs compared to CQL, but this is acceptable given its performance improvement.**
>
> We conducted experiments on the hfctah-med-rep task to evaluate the computational costs of different methods. The results are shown below:
> | Method | Total Time | Seconds per Epoch (1000 epochs)                                  |
> | ------ | ---------- | ---------------------------------------------------------------- |
> | Madoc  | 5 hours    | 1.8s for grouping (200 epochs), 14s for  calibration, 4s for SAC |
> | CQL    | 2 hours    | 7s for CQL                                                       |
> | MOREC  | 6 hours    | 5s for dynamics reward function, 16s for policy training         |
>
> The table indicates that Madoc involves three training stages and consumes more computational cost compared to CQL. However, given the significant performance improvement, this additional computational cost is justifiable. We will also include this discussion in the revised appendix.
> ### Q5: How good is the reward model?
> **The reward model can accurately reflect the dynamics gap under different parameters.**
>
> To verify the role of the reward model, we designed an experiment to test whether it can act as a scoring function to evaluate the truthfulness of transitions. As shown **in the right half of the PDF of the "global" response**, we stored the checkpoint of the trained reward model and used it to test the reward results (the reward range should be [-4, 4]) it outputs under different dynamics.
> In this simple test, we only changed two parameters: the gravity coefficient and body_mass_1 (the target values are -9.81 and 6.36, respectively). We sampled 256 transitions with the behavior policy under each parameter condition to calculate the mean reward. The results, shown in the heat map, indicate that when the domain parameters are closer to the target parameters, the output reward is higher. This suggests that it can accurately reflect the dynamics gap under different parameters.
> ### Q6: Comparison with independent learning methods.
> **Independent SAC agents perform worse than Madoc agents in domain calibration.**
>
> Independent SAC (ISAC) treats each agent as an independent individual, optimizing each policy with shared rewards without considering the joint policy. We conducted relevant experiments and the results are shown below:
> | Task           | Madoc    | ISAC      |
> | -------------- | -------- | --------- |
> | hfctah-med     | 91.9±7.7 | 81.1±14.7 |
> | hfctah-med-rep | 95.7±9.9 | 75.0±13.1 |
> | hfctah-med-exp | 96.9±5.3 | 75.2±16.1 |
>
> The results indicate that ISAC performs worse than Madoc in domain calibration. We believe the main reason is that all physics parameters are interrelated and cooperative. The independent learning method ignores the policy changes of other calibration agents, leading to non-stationary problems [2]. IPPO [3] performs well on some SMAC tasks because these tasks do not have high requirements for cooperativeness.
>
> Reference:
>
> [1] Data-Efficient Pipeline for Offline Reinforcement Learning with Limited Data. NeurIPS, 2022.
>
> [2] Dealing with Non-Stationarity in Multi-Agent Deep Reinforcement Learning. arXiv, 2019.
>
> [3] Is Independent Learning All You Need in the StarCraft Multi-Agent Challenge? arXiv, 2020.

---

> > ### Comment · Reviewer_Cdsr · 2024-08-10
> >
> > Thanks for the detailed response. The responses addressed all of my concerns and integrating them to the paper could improve the paper. I still keep my score and support acceptance as the methodology of the paper is both novel and well communicated in my opinion.

---

### Comment · Area_Chair_crr9 · 2024-07-31
**Should have another review in by tomorrow**

I've recruited an emergency reviewer who should finish by tomorrow at the latest.
Apologies for the lateness.

---

### Author Rebuttal · Authors · 2024-08-07

We appreciate valuable comments from all reviewers. We have carefully clarified the ambiguous parts and supplemented our work with additional experiments to address the raised issues. Our revisions can be briefly summarized as follows:

- Method.
    - We have further elaborated on the motivation and derivation process of the optimization objective, supplemented the explanations of relevant definitions, and improved the representation to make it easier to understand.
    - The supplementary PDF presents the revised version of Fig. 2.
- Experiments.
    - We have provided a more detailed explanation of the empirical results and supplemented them with relevant experiments to demonstrate the superiority of our method.
    - We have added the comparison results with new baseline algorithms in some scenarios, highlighting the differences and demonstrating the superiority of our method.
    - We have presented the comparison of model complexity and computational cost with offline algorithms, reflecting the feasibility of our method.

The supplementary PDF consists of two parts:
- The left half presents the revised version of Fig. 2 from the manuscript, with corrected errors in the legend. Thank you for pointing out this crucial typo, which has helped us improve our work.
- The right half contains the verification of the reward model, demonstrating that it can reflect the dynamics gap under different physics parameters. The complete description can be found in Q5 for Reviewer Cdsr.

We hope that our responses address all the questions and concerns. If we have missed anything, please let us know. We are always willing to resolve any further issues and look forward to the ensuing insightful discussions.

---

### Decision · Program_Chairs · 2024-09-25

**Decision:**

Accept (poster)

**Comment:**

This paper frames domain calibration as a cooperative Multi-Agent Reinforcement Learning (MARL) problem, and finds significant performance gains in smaller data regimes than tackled in previous works.  Reviewers originally raised a number of concerns regarding the cost of the method, clarity, the motivation for such an approach, and the strength compared to existing alternatives in the literature.  Ultimately, the authors were able to address these points satisfactorily in the discussion, and I’m happy to recommend acceptance on the basis of the strong empirical results and novelty of the framework.